# Continual World: A Robotic Benchmark For Continual Reinforcement Learning

**Maciej Wołczyk**\*
Jagiellonian University
Kraków, Poland
maciej.wolczyk@doctoral.uj.edu.pl

**Michał Zając**\*
Jagiellonian University
Kraków, Poland
emzajac@gmail.com

**Razvan Pascanu**
DeepMind
London, UK
razp@google.com

**Łukasz Kuciński**
Polish Academy of Sciences
Warsaw, Poland
lkucinski@impan.pl

**Piotr Miłoś**
Polish Academy of Sciences,
University of Oxford,
deepsense.ai
Warsaw, Poland
pmilos@impan.pl

## Abstract

Continual learning (CL) — the ability to continuously learn, building on previously acquired knowledge — is a natural requirement for long-lived autonomous reinforcement learning (RL) agents. While building such agents, one needs to balance opposing desiderata, such as constraints on capacity and compute, the ability to not catastrophically forget, and to exhibit positive transfer on new tasks. Understanding the right trade-off is conceptually and computationally challenging, which we argue has led the community to overly focus on *catastrophic forgetting*. In response to these issues, we advocate for the need to prioritize forward transfer and propose *Continual World*, a benchmark consisting of realistic and meaningfully diverse robotic tasks built on top of Meta-World [54] as a testbed. Following an in-depth empirical evaluation of existing CL methods, we pinpoint their limitations and highlight unique algorithmic challenges in the RL setting. Our benchmark aims to provide a meaningful and computationally inexpensive challenge for the community and thus help better understand the performance of existing and future solutions. Information about the benchmark, including the open-source code, is available at https://sites.google.com/view/continualworld.

## 1 Introduction

Change is ubiquitous. Unsurprisingly, due to evolutionary pressure, humans can quickly adapt and creatively reuse their previous experiences. In contrast, although biologically inspired, deep learning (DL) models excel mostly in static domains that satisfy the i.i.d. assumption, as for example in image processing [28, 49, 10, 40], language modelling [52, 11] or biological applications [47]. As the systems are scaled up and deployed in open-ended settings, such assumptions are increasingly questionable; imagine, for example, a robot that needs to adapt to the changing environment and the wear-and-tear of its hardware. *Continual learning* (CL), an area that explicitly focuses on such problems, has been gaining more attention recently. The progress in this area could offer enormous advantages for deep neural networks [19] and move the community closer to the long-term goal of building intelligent machines [20].

---

\*equal contribution

35th Conference on Neural Information Processing Systems (NeurIPS 2021).

Evaluation of CL methods is challenging. Due to the sequential nature of the problem that disallows parallel computation, evaluation tends to be expensive, which has biased the community to focus on toy tasks. These are mostly in the domain of supervised learning, often relying on MNIST. In this work, we expand on previous discussions on the topic [45, 16, 30, 46] and introduce a new benchmark, *Continual World*. The benchmark is built on realistic robotic manipulation tasks from Meta-World [54], benefiting from its diversity but also being computationally cheap. Moreover, we provide shorter auxiliary sequences, all of which enable a quick research cycle. On the conceptual level, a fundamental difficulty of evaluating CL algorithms comes from the different desiderata for a CL solution. These objectives are often opposing each other, forcing practitioners to explicitly or implicitly make trade-offs in their algorithmic design that are data-dependent. *Continual World* provides more meaningful relationships between tasks, answering recent calls [19] to increase attention on forward transfer.

Additionally, we provide an extensive evaluation of a spectrum of commonly used CL methods. It highlights that many approaches can deal relatively well with *catastrophic forgetting* at the expense of other desiderata, in particular forward transfer. This emphasizes our call for focusing on *forward transfer* and the need for more benchmarks that allow for common structure among the tasks.

The main contribution of this work is a CL benchmark that poses optimizing forward transfer as the central goal and shows that existing methods struggle to outperform simple baselines in terms of the forward transfer capability. We release the code[2] both for the benchmark and 7 CL methods, which aims to provide the community helpful tools to better understand the performance of existing and future solutions. We encourage to visit the website[3] of the project and participate in the Continual World Challenge.

## 2   Related work

The field of continual learning has grown considerably in the past years, with numerous works forming new subfields [23] and finding novel applications [48]. For brevity, we focus only on the papers proposing RL-based benchmarks and point to selected surveys of the entire field. [19] provide a high-level overview of CL and argue that learning in a non-stationary setting is a fundamental problem for the development of AI, highlighting the frequent connections to neuroscience. On the other hand, [13, 38] focus on describing, evaluating, and relating CL methods to each other, providing a taxonomy of CL solutions that we use in this work.

The possibility of applying CL methods in reinforcement learning scenarios has been explored for a long time, see [25] for a recent review. However, no benchmark has been widely accepted by the community so far, which is the aim of this work. Below we discuss various benchmarks and environments considered in the literature.

**Supervised settings** MNIST has been widely used to benchmark CL algorithms in two forms [27]. In the permuted MNIST, the pixels of images are randomly permuted to form new tasks. In the split MNIST, tasks are defined by classifying non-overlapping subsets of classes, e.g. 0 vs. 1 followed by 2 vs. 3. A similar procedure has been applied to various image classification tasks like CIFAR-10, CIFAR-100, Omniglot or mini-ImageNet [2, 46, 5]. Another benchmark is CORe50 [31], a dataset for continuous object recognition. Recent work [29] proposes a benchmark based on language modeling. We find that many of these benchmarks are challenging and allow to measure forgetting. However, we argue they are not geared towards measuring forward transfer or for highlighting important RL-specific characteristics of the CL problem.

**Atari** The Atari 2600 suite [8] is a widely accepted RL benchmark. Sequences of different Atari games have been used for evaluating continual learning approaches [43, 27]. Using Atari can be computationally expensive, e.g., training a sequence of ten games typically requires 100M steps or more. More importantly, as [43] notes, these games lack a meaningful overlap, limiting their relevance for studying transfers. **Continuous control** [32, 24] use continuous control tasks such as Humanoid or Walker2D. However, the considered sequences are short, and the range of experiments is limited. [34] use Meta-World tasks, similarly to us, for evaluations of their continual learning method, but the work is not aimed at building a benchmark. As such, it uses the Meta-World's MT10 preset and

---

[2] `https://github.com/awarelab/continual_world`
[3] `https://sites.google.com/view/continualworld/home`

does not provide an in-depth analysis of the tasks or other CL methods. **Maze navigation** A set of 3D maze environments is used in [43]. The map structure and objects that the agent needs to collect change between tasks. It is not clear, though, if the tasks provide enough diversity. [30] propose CRLMaze, 3D navigation scenarios for continual learning, which solely concentrate on changes of the visual aspects. **StarCraft** [45] present a StarCraft campaign (11 tasks) to evaluate a high-level transfer of skills. The main drawback of this benchmark is excessive computational demand (often more than 1B frames). **Minecraft** [50] propose simple scenarios within the Minecraft domain along with a hierarchical learning method. The authors phrase the problem as lifelong learning and do not use typical CL methods. **Lifelong Hanabi** [36] consider a multi-agent reinforcement learning setting based on Hanabi, a cooperative game requiring significant coordination between agents. On the other hand, we focus on the single agent setting with changing environment, which allows us to bypass the computational complexity needed to model interactions between agents and highlight issues connected to learning in a changing world. **Causal World** [3] propose an environment for robotic manipulation tasks which share causal structure. Although they investigate issues deeply connected to learning in a changing world, such as generalization to new tasks and curricula, they do not directly consider continual learning. **Jelly Bean World** [39] provide interesting procedurally generated grid world environments. The suite is configurable and can host a non-stationary setting. It is unclear, however, if such environments reflect the characteristics of real-world challenges.

## 3  Continual learning background

Continual learning (CL) is an area of research which focuses on building algorithms capable of handling non-stationarity. They should be able to sequentially acquire new skills and solve novel tasks without forgetting the previous ones. Such systems are desired to accommodate over extended periods swiftly, which is often compared to human capabilities and alternatively dubbed as lifelong learning. CL is intimately related to multi-task learning, curriculum learning, meta-learning, with some key differences. Multi-task assumes constant access to all tasks, thus ignoring non-stationarity. Curriculum learning focuses on controlling the task ordering and often the learning time-span. Meta-learning, a large field of its own, sets the objective to develop procedures that allow fast adaptation within a task distribution and usually ignores the issue of non-stationarity.

The CL objective is operationalized by the training and evaluation protocols. The former typically consists of a sequence of tasks (their boundaries might be implicit and smooth). The latter usually involves measuring *catastrophic forgetting*, *forward transfer*, and *backward transfer*. The learning system might also have constrained resources: *computations*, *memory*, *size of neural networks*, and the *volume of data samples*. A fundamental observation is that the above aspects and desiderata are conflicting. For example, given unlimited resources, one might mitigate forgetting simply by storing everything in memory and paying a high computational cost of rehearsing all samples from the past.

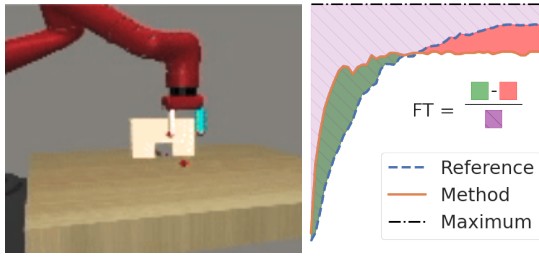

Figure 1: Left graph shows task PEG-UNPLUG-SIDE-V1 and the right graph presents forward transfer from SHELF-PLACE-V1 to PEG-UNPLUG-SIDE-V1. In this case $FT = 0.10$.

Another pair of objectives that are problematic for current methods are forgetting and forward transfer. For neural networks, existing methods propose to limit network plasticity. These alleviate the problem of forgetting, however, at the cost of choking the further learning process. We advocate for more nuanced approaches. Importantly, to make the transfer possible, our benchmark is composed of related tasks. We also put modest bounds on resources. This requirement is in line with realistic scenarios, demanding computationally efficient adaptation and inference. In a broader sense, we hope to address a data efficiency challenge, one of the most significant limitations of the current deep (reinforcement) learning methods. We conjecture that forward transfer might greatly improve the situation and possibly one day enable us to create systems with human-level cognition capabilities, in line with similar thoughts expressed in [19].

# 4 Continual World benchmark

Continual World is a new benchmark designed to be a testbed for evaluating RL agents on the challenges advocated by the CL paradigm, described in Section 3, as well as highlighting the RL-specific algorithmic challenges for CL (see Section 6.1). As such it is aimed at being valuable to both the CL and RL communities. Continual World consists of realistic robotic manipulation tasks, aligned in a sequence to enable the study of forward transfer. It is designed to be challenging while computationally accessible.[4] The benchmark is based on Meta-World, a suite of robotic tasks already established in the community. This enables easy comparisons with the related fields of multi-task and meta-learning reinforcement learning, potentially highlighting one benefit of CL framing, namely that of dealing with different reward scales as we discuss more in detail in Appendix H. Continual World comes with open-source code that allows for easy development and testing of new algorithms and provides implementations of 7 existing algorithms. Finally, it allows highlighting RL-specific challenges for the CL setting. We believe that our work is a step in the right direction towards reliable benchmarks of CL. We realize, however, that it will need to evolve as the field progresses. We leave a discussion on future directions and limitations to Section 4.4.

## 4.1 Metrics

To facilitate further discussion, we start with defining metrics. These are rather standard in the CL setting [42]. Assume $p_i(t) \in [0, 1]$ to be the performance (success rate) of task $i$ at time $t$. As a measure of performance, we take the average success rate of achieving a goal specified by a given task when using randomized initial conditions and stochastic policies (see also Section 4.3).[5] Each task is trained for $\Delta = 1M$ steps. The main sequence has $N = 20$ tasks and the total sample budget is $T = N \cdot \Delta = 20M$. The $i$-th task is trained during the interval $t \in [(i - 1) \cdot \Delta, i \cdot \Delta]$. We report the following metrics:

**Average performance**. The average performance at time $t$ is (see Figure 3)

$$P(t) := \frac{1}{N} \sum_{i=1}^{N} p_i(t). \tag{1}$$

Its final value, $P(T)$, is a traditional metric used in the CL research. This is the objective we use for tuning hyperparameters. We have $P(t) \in [0, 1]$ for each $t$.

**Forward transfer**. We measure the forward transfer of a method as the normalized area between its training curve and the training curve of the reference, single-task, experiment, see Figure 1. Let $p_i^b \in [0, 1]$ be the reference performance[6] then the forward transfer for the task $i$, denoted by $\text{FT}_i$, is

$$\text{FT}_i := \frac{\text{AUC}_i - \text{AUC}_i^b}{1 - \text{AUC}_i^b}, \quad \text{AUC}_i := \frac{1}{\Delta} \int_{(i-1)\cdot\Delta}^{i\cdot\Delta} p_i(t) \mathrm{d}t, \quad \text{AUC}_i^b := \frac{1}{\Delta} \int_{0}^{\Delta} p_i^b(t) \mathrm{d}t,$$

The average forward transfer for all tasks, FT, is defined as

$$\text{FT} = \frac{1}{N} \sum_{i=1}^{N} \text{FT}_i. \tag{2}$$

We note that $\text{FT}_i \le 1$ and they might be negative. In our experiments, we also measure backward transfer. As it is negligible, see Appendix E.1.

**Forgetting**. For task $i$, we measure the decrease of performance after ending its training, i.e.
$$F_i = p_i(i \cdot \Delta) - p_i(T). \tag{3}$$

Similarly to FT, we report $F = \frac{1}{N} \sum_{i=1}^{N} F_i$. We have $F_i \le 1$ for any $i$ and consequently FT $\le 1$. It is possible that $F_i$ are negative, which would indicate backward transfer. We do not observe this in practice, see Appendix E.1.

---

[4]We use 8-core machines without GPU. Training the CW20 sequence of twenty tasks takes about 100 hours. We also provide shorter 10 and 3 task sequences to speed up the experimental loop further.

[5]Stochastic evaluations are slightly more smooth and have little difference to the deterministic ones.

[6]Note that we avoid trivial tasks, for which $\text{AUC}_i^b = 1$ is making the metric ill-defined. Additionally, we acknowledge the dependency on the hyperparameters of the learning algorithm and that there are alternative quantities of interest like relative improvement in performance rather than faster learning.

## 4.2 Continual World tasks

This section describes the composition of Continual World benchmark and the rationale behind its design. We decided to base on Meta-World [54], a fairly new but already established robotic benchmark for multi-task and meta reinforcement learning. From a practical standpoint, Meta-World utilizes the open-source MuJoCo physics engine [51], prized for speed and accuracy. Meta-World provides 50 distinct manipulation tasks with everyday objects using a simulated robotic Sawyer arm. Although the tasks vary significantly, the structure and semantics of observation and action spaces remain the same, allowing for transfer between tasks. Each observation is a 12-dimensional vector containing $(x, y, z)$ coordinates of the robot's gripper and objects of interest in the scene. The 4-dimensional action space describes the direction of the arm's movement in the next step and the gripper actuator delta. Reward functions are shaped to make each task solvable. In evaluations, we use a binary *success metric* based on the distance of the task-relevant object to its goal position. This metric is interpretable and enables comparisons between tasks. For more details about the rewards and evaluation metrics, see [54, Section 4.2, Section 4.3].

**CW20, CW10, triplets sequences** The core of our benchmark is CW20 sequence. Out of 50 tasks defined in Meta-World, we picked those that are not too easy or too hard in the assumed sample budget $\Delta = 1M$. Aiming to strike a balance between the difficulty of the benchmark and computational requirements, we selected 10 tasks. The tasks and their ordering were based on the transfer matrix (see the next paragraph), so that there is a high variation of forward transfers (both in the whole list and locally). We refer to these ordered tasks as CW10, and CW20 is CW10 repeated twice. We recommend using CW20 for final evaluation; however, CW10 is already very informative in most cases. Due to brevity constraints, we present an ablation with an alternative ordering of the tasks and a longer sequence of 30 tasks in Appendix G, however, these experiments do not alter our findings. Additionally, to facilitate a fast development cycle, we propose a set of triplets, sequences of three tasks which exhibit interesting learning dynamics.

The CW10 sequence is: HAMMER-V1, PUSH-WALL-V1, FAUCET-CLOSE-V1, PUSH-BACK-V1, STICK-PULL-V1, HANDLE-PRESS-SIDE-V1, PUSH-V1, SHELF-PLACE-V1, WINDOW-CLOSE-V1, PEG-UNPLUG-SIDE-V1.

**Transfer matrix** Generally, the relationship between tasks and its impact on learning dynamics of neural networks is hard to quantify, where semantic similarity does not typically lead to transfer [14]. To this end, we consider a minimal setting, in which we finetune on task $t_2$ a model pretrained on $t_1$, using the same protocol as the benchmark (e.g., different output heads, see Section 4.3). This provides neural network-centric insight into the relationship between tasks summarized in Figure 2, and allows us to measure *low-level transfer* between tasks, i.e., the ability of the model to reuse previously acquired features. See Appendix D for more results and extended discussion.

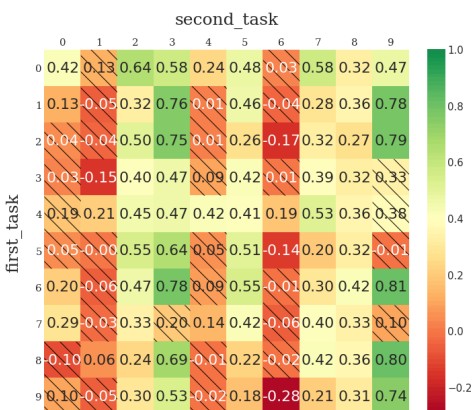

Figure 2: Transfer matrix, see Section 4.2. Each cell represents the forward transfer from the first task to the second one. We shaded the cells for which 0 belongs to their 90% confidence interval.

Notice that there are only a few negative forward transfer cases, and those are of a rather small magnitude (perhaps unsurprisingly, as the tasks are related). There are also visible patterns in the matrix. For instance, some tasks such as PEG-UNPLUG-SIDE-V1 or PUSH-BACK-V1 benefit from a relatively large forward transfer, (almost) irrespective of the first task. Furthermore, the average forward transfer given the second task (columns) is more variable than the corresponding quantity for the first task (rows).

Note that some transfers on the diagonal (i.e., between the same tasks) are relatively small. We made a detailed analysis of possible reasons, which revealed that the biggest negative impact is due to the replay buffer resets, which seems, however, unavoidable for off-diagonal cases, see Section 6.1 for details.

Importantly, we use this matrix to estimate what level of forward transfer a good CL method should be able to achieve. We expect that a model which is able to remember all meaningful

aspects of previously seen tasks would transfer at least as well as if one were just fine-tuning after learning the best choice between the previous tasks. For a sequence $t_1, \ldots, t_N$ we set the reference forward transfer, RT, to be

$$\text{RT} := \frac{1}{N} \sum_{i=2}^{N} \max_{j<i} \text{FT}(t_j, t_i), \tag{4}$$

where $\text{FT}(t_j, t_i)$ is the transfer matrix value for $t_j, t_i$. For the CW20 sequence, the value is $\text{RT} = 0.46$. Note that a *model can do better* that this by composing knowledge from multiple previous tasks.

## 4.3 Training and evaluation details

We adapt the standard Meta-World setting to CL needs. First, we use separate policy heads for each task, instead of the original one-hot task ID inputs (we provide ablation experiments for this choice in Appendix G). Second, in each episode, we randomize the positions of objects in the scene to encourage learning more robust policies. We use an MLP network with 4 layers of 256 neurons.

For training, we use soft actor-critic (SAC) [17], a popular and efficient RL method for continuous domains. SAC is an off-policy algorithm using replay buffer, which is an important aspect for CL, particularly for methods relying on rehearsing old trajectories. SAC is based on the so-called maximum entropy principle; this results in policies which explore better and are more robust to changes in the environment dynamics. Both of these qualities might be beneficial in CL.

We note that the size of the neural network and optimization details of the SAC algorithm (like batch size) put constraints on "the amount of compute". Intentionally, these are rather modest, which is in line with CL desiderata, see Section 3. Similarly, we limit the number of timesteps to $1M$, which is a humble amount for modern-day deep reinforcement learning. We picked tasks to be challenging but not impossible within this budget. We note that training in the RL setting tends to be less stable than in the supervised one. We recommend using multiple seeds, in our experiments, we typically used 20 and calculate confidence intervals; we used the bootstrap method. We choose hyperparameters that maximize average performance (1). In our experiments, we tune common parameters for SAC and the method-specific hyperparameters separately. All details of the training and evaluation setup are presented in Appendix A.

## 4.4 Limitations of Continual World

As any benchmark, we are fully aware that ours will not cover the entire spectrum of problems that one might be interested in. Here we summarize a few limitations that we hope to overcome in a future instantiation of this benchmark:

**Input space** We use a small 12 dimensional observation space. This is key to achieve modest computational demand. However, richer inputs could allow for potentially more interesting forms of transfer (e.g., based on visual similarity of objects) and would allow inferring the task from the observation, which is currently impossible.

**Reliance on SAC** We use the SAC algorithm [17], which is considered a standard choice for continuous robotic tasks. However, there is a potential risk of overfitting to the particularities of this algorithm and exploring alternative RL algorithms is important.

**Task boundaries** We rely on task boundaries. One can rely on task inference mechanisms (e.g. [35, 41]) to resolve this limitation, though we acknowledge the importance to extend the benchmark towards allowing and testing for task inference capabilities. Also, testing for algorithms dealing with continuous distributional drift is not possible in the current format.

**Output heads** We rely on using a separate head for each new task, similar to many works on continual learning. We opt for this variant based on its simplicity and better performance than using one-hot encoding to indicate a task. We believe that the lack of semantics of the one-hot encoding would further impede transfer, as the relationship between tasks can not be inferred. We carry ablation studies with using one-hot encoding as an input and a single head architecture, a setting that is already compatible with our benchmark. We regard this aspect as an important future work, and in particular, we are exploring alternative encoding of input to make this choice more natural. A coherent domain, like Continual World, provides a unique opportunity to exploit a consistent output layer as its semantics does not change between tasks.

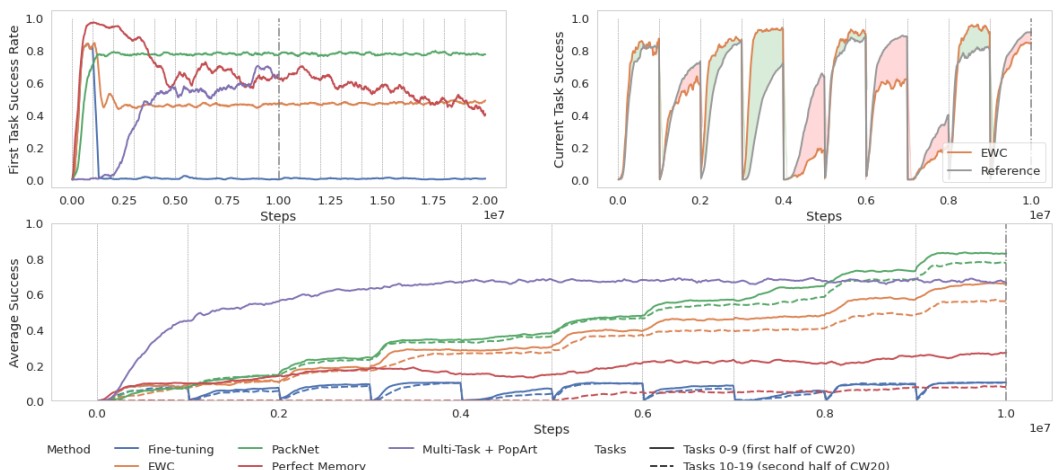

Figure 3: Training curves for selected CL methods and multi-task. The upper left panel shows the performance on the first task for a subset of methods throughout the whole training. Note that due to the use of different output heads, we do not see a second bump when revisiting this task at time $10M$. The upper right panel shows the performance on the current task being trained for EWC compared to a reference (a model learning only that task from scratch). The bottom plot shows the average performance. Solid lines show the performance of the model training on the first 10 tasks (where 1 means being able to solve all of them). Dashed lines show the performance of learning the same tasks in the second half of the benchmark. Note that dashed lower are below solid ones, indicating lower performance on the second pass, even if the agent has already previously learned the tasks and has access to relevant features.

**The difficulty and number of tasks** The number of tasks is relatively small. CW20, the main sequence we use, consists of only 10 different tasks, which are then repeated. We believe the repetition of tasks is important for a CL benchmark, leading to interesting observations. We check also that results are quantitatively similar on a sequence of 30 tasks, see Appendix G. However, longer sequences, potentially unbounded, are needed to understand the various limitations of existing algorithms. For example, the importance of *graceful forgetting* or dealing with systems that run out of capacity, a scenario where *there is no multi-task solution for the sequence of observed tasks*. This is particularly of interest for methods such as PackNet [33]. Additionally, we provide the number of tasks in advance. Dealing with an unknown number of tasks might raise further interesting questions. Finally, in future iterations of the benchmark, it is important to consider more complex tasks or more complex relationships between tasks to remain a challenge to existing methods. Our goal was to provide a benchmark that is approachable by existing methods, as not to stifle progress.

**Low-level transfer** We focus on low-level transfers via neural network features and weights. As such, we do not explicitly explore the ability of the learning process to exploit the compositionality of behavior or to rely on a more interesting semantic level. While we believe such research is crucial, we argue that solving low-level transfer is equally important and might be a prerequisite. So, for now, it is beyond the scope of this work, though future iterations of the benchmark could contain such scenarios.

## 5 Methods

We now sketch 7 CL methods evaluated on our benchmark. Some of them were developed for RL, while others were meant for the supervised learning context and required non-trivial adaptation. We aimed to cover different families of methods; following [13], we consider three classes: regularization-based, parameter isolation and replay methods. An extended description and discussion of these methods are provided in Appendix B.

**Regularization-based Methods** This family builds on the observation that one can reduce forgetting by protecting parameters that are important for the previous tasks. The most basic approach often

dubbed **L2** [27] simply adds a $L_2$ penalty, which regularizes the network not to stray away from the previously learned weights. In this approach, each parameter is equally important. **Elastic Weight Consolidation (EWC)** [27] uses the Fisher information matrix to approximate the importance of each weight. **Memory-Aware Synapses (MAS)** [4] also utilizes a weighted penalty, but the importance is obtained by approximating the impact each parameter has on the output of the network. **Variational Continual Learning (VCL)**, follows a similar path but uses variational inference to minimize the Kullback-Leibler divergence between the current distribution of parameters (posterior) and the distribution for the previous tasks (prior).

**Parameter Isolation Methods** This family (also called modularity-based) forbids any changes to parameters that are important for the previous tasks. It may be considered as a "hard" equivalent of regularization-based methods. **PackNet** [33] "packs" multiple tasks into a single network by iteratively pruning, freezing, and retraining parts of the network at task change. PackNet is closely related to progressive neural networks [43], developed in the RL context.

**Replay Methods** Methods of this family keep some samples from the previous tasks and use them for training or as constraints to reduce forgetting. We use a **Perfect Memory** baseline, a modification of our setting which remembers all the samples from the past (i.e., without resetting the buffer at the task change). We also implemented **Averaged Gradient Episodic Memory (A-GEM)** [12], which projects gradients from new samples as to not interfere with previous tasks. We find that A-GEM does not perform well on our benchmark.

**Multi-task learning** In multi-task learning, a field closely related to CL, tasks are trained simultaneously. By its design, it does not suffer from forgetting, however, it is considered to be hard as multiple tasks "compete for the attention of a single learning system", see [21, 44]. We find that using reward normalization as in PopArt [21] is essential to achieve good performance. See Appendix H.

# 6 Experiments

Now we present empirical results; these are evaluations of a set of 7 representative CL methods (as described in Section 5) on our Continual World benchmark. We focus on *forgetting and transfers* while keeping fixed constraints on computation, memory, number of samples, and neural network architecture. Our main empirical contributions are experiments on the long CW20 sequence and following high-level conclusions. For a summary see Table 1, Figure 3 and for an extensive discussion, we refer to Appendix E (including results for the shorter sequence, CW10). In Appendix G we provide various ablations and detailed analysis of sensitivity to the CL-specific hyperparameters.

**Performance** The performance (success rate) averaged over tasks (eq. (1)) is a typical metric for the CL setting. Pack-Net seems to outperform other methods, approaching 0.8 from the maximum of 1.0, outperforming multi-task solutions which might struggle with different reward scales, a problem elegantly avoided in the CL framing. Other methods perform considerably worse. A-GEM and Perfect Memory struggle. We further discuss possible reasons in Section 6.1.

| method | performance | forgetting | f. transfer |
|---|---|---|---|
| **Fine-tuning** | 0.05 [0.05, 0.06] | 0.73 [0.72, 0.75] | **0.20 [0.17, 0.23]** |
| **L2** | 0.43 [0.39, 0.47] | 0.02 [0.00, 0.03] | -0.71 [-0.87, -0.57] |
| **EWC** | 0.60 [0.57, 0.64] | 0.02 [-0.00, 0.05] | -0.17 [-0.24, -0.11] |
| **MAS** | 0.51 [0.49, 0.53] | 0.00 [-0.01, 0.02] | -0.52 [-0.59, -0.47] |
| **VCL** | 0.48 [0.46, 0.50] | 0.01 [-0.01, 0.02] | -0.49 [-0.57, -0.42] |
| **PackNet** | **0.80 [0.79, 0.82]** | 0.00 [-0.01, 0.01] | 0.19 [0.15, 0.23] |
| **Perfect Memory** | 0.12 [0.09, 0.15] | 0.07 [0.05, 0.10] | -1.34 [-1.42, -1.27] |
| **A-GEM** | 0.07 [0.06, 0.08] | 0.71 [0.70, 0.73] | 0.13 [0.10, 0.16] |
| **MT** | 0.51 [0.48, 0.53] | — | — |
| **MT (PopArt)** | 0.65 [0.63, 0.67] | — | — |
| **RT** | — | — | **0.46** |

Table 1: Results on CW20, for CL methods and multi-task training. Metrics are defined in Section 4.1. RT is eq. (4). We used 20 seeds and provide 90% confidence intervals.

**Forgetting** We observe that most CL methods are usually efficient in mitigating forgetting. However, we did not notice any boost when revisiting a task (see Figure 3). Even if a different output head was employed, relearning the internal representation should have had an impact unless it changed considerably when revisiting the task. Additionally, we found A-GEM difficult to tune; consequently, with the best hyperparameter settings, it is relatively similar to the baseline fine-tuning method (see details in Appendix C).

**Transfers** For all methods, forward transfer for the second ten tasks (and the same tasks are revisited) drops compared to the first ten tasks. This is in stark contrast to forgetting, which seems to be well under control. Among all methods, only fine-tuning and PackNet are able to achieve positive forward transfer (0.20 and 0.19, resp.) as well as on the first (0.32 and 0.21, resp.) and the second (0.08 and 0.17, resp.) half of tasks. However, these are considerably smaller than RT = 0.46, which in principle can even be exceeded, and which should be reached by a model that remembers all meaningful aspects of previously seen tasks, see (4). These results paint a fairly grim picture: we would expect improvement, rather than deterioration in performance, when revisiting previously seen tasks. There could be multiple reasons for this state of affairs. It could be attributed to the loss of plasticity, similar to the effect observed in [6]. Another reason could be related to the interference between CL mechanisms or setting and RL, for instance, hindering exploration. We did not observe any substantial cases of **backward transfer**, even though the benchmark is well suited to study this question due to the revisiting of tasks. See Appendix E.1.

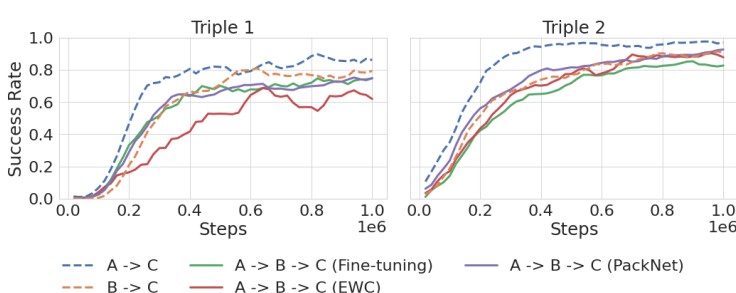

Figure 4: **How forgetting impacts the forward transfer.** Two different triplets of tasks learnt in sequence. An ideal agent learning on a sequence $A \rightarrow B \rightarrow C$ should have at least as good performance on task $C$ as an agent which just learns $A \rightarrow C$. In reality, an interfering task $B$ reduces this transfer, even when continual learning approaches are used.

**Triplets experiments** We illustrate how forgetting and forward transfer interact with each other in a simpler setting of three task sequences, see Figure 4 and Appendix F. We focus on sequences of tasks $A \rightarrow B \rightarrow C$, where $A \rightarrow C$ has significant positive forward transfer and $B \rightarrow C$ has a smaller or even negative transfer. An efficient CL agent should be able to use information from $A$ to get good performance on $C$. However, interference introduced by $B$ reduces the final forward transfer (see Figure 4). The drive for reducing forgetting in CL agents has been primarily to perform well on previous tasks when we revisit them. With this example, we argue that an equally important reason to improve the memory of CL agents is to efficiently use past experiences to learn faster on new tasks. Currently, the tested CL methods often are not able to outperform the forgetful fine-tuning baseline. Observe that even the modularity-based PackNet approach struggles with this task. This possibly indicates that using the activation mask from task $B$ is enough to deteriorate the performance.

**PackNet** PackNet stands out in our evaluations. We conjecture that developing related methods might be a promising research direction. Besides further increasing performance, one could mitigate the limitations of PackNet. PackNet relies on knowing task identity during evaluation. While this assumption is met in our benchmark, it is an interesting topic for future research to develop methods that cope without task identity. Another nuisance is that PackNet assigns some fixed fraction of parameters to a task. This necessitates knowledge of the length of the sequence in advance. Additionally, when the second ten tasks of CW20 start, PackNet performance degrades, showing its potentially inefficient use of capacity and past knowledge, given that the second ten tasks are identical with the first ten and hence no additional capacity is needed. In a broader context, we speculate that parameter isolation methods might be a promising direction towards better CL methods.

**Resources usage** In practical applications it is important to consider resources usage. All tested methods have relatively small overheads. For example, PackNet needs only 15% more time than the baseline fine-tuning and it requires 50% more neural network parameters (which is negligible when small networks like ours). See Appendix B.4 for details concerning other methods.

**Other observations** In stark contrast with the supervised learning setting, we found that replay based methods (Perfect Memory and A-GEM) suffer from poor performance. This is even though we allow for a generous replay, which could store the whole experience. Explaining and amending this situation is, in our view, an important research question. We conjecture that this happens due to the regularization of the critic network (which was unavoidable for these methods). We found multi-task

learning attaining lower scores than PackNet, the best CL method and comparable to the second one, EWC. We think this suggests interesting research directions for multi-task learning.

## 6.1 RL-Related Challenges

Reinforcement learning brings a set of issues not present in the supervised learning setting, e.g., exploration, varying reward scales, and stochasticity of environments. We argue that it is imperative to have a reliable benchmark to assess the efficiency of CL algorithms with respect to these problems. We find that some current methods are not well adjusted to the RL setting and require non-trivial conceptual considerations and careful tuning of hyperparameters, see details in Appendix C.

An important design choice is whether or not to regularize the critic in the actor-critic framework (e.g. in SAC). We find it beneficial to focus on reducing forgetting in the actor while allowing the critic to freely adapt to the current task (note that critic is used only in training of the current task), similar to [46]. On the other hand, a forgetful critic is controversial. This can be sharply seen when the same task is repeated and the critic needs to learn from scratch. Additionally, not all methods can be trivially adapted to the 'actor-only regularization' setting, as for example replay based methods. In Appendix C we examine these issues empirically, by showing experiments with critic regularization for EWC.

Another aspect is the exploration and its non-trivial impact on transfers. As it was observed, transfers from a given task to the same one are sometimes poor. We show in Appendix D.1 that this results from the fact that at the task change the replay buffer is emptied and SAC collects new samples from scratch, usually by using the uniform policy. Learning on these random samples reduces performance on the current task and thus the forward transfer. Experimentally, we find that not resetting the buffer or using the current policy for exploration improves the transfer on the diagonal.

# 7 Conclusions and Future Work

In this work, we present Continual World, a continual reinforcement learning benchmark, and an in-depth analysis of how existing methods perform on it. The benchmark is aimed at facilitating and standardizing the CL system evaluation, and as such, is released with code, including implementation of 7 representative CL algorithms. We argue for more attention to *forward transfer* and the interaction between forgetting and transfer, as many existing methods seem to sacrifice transfer to alleviate forgetting. In our opinion, this should not be the aim of CL, and we need to strike a different balance between these objectives.

We made several observations, both conceptual and empirical, which open future research directions. In particular, we conjecture that parameter isolation methods are a promising direction. Further, we identified a set of critical issues at the intersection of RL and CL. Resolving critic regularization and efficient use of multi-task replays seem to be the most pressing ones. Our benchmark highlights some challenges, which in our view are relevant and tangible now. In the long horizon, achieving high-level transfers, removing task boundaries, and scaling up are among significant goals for future editions of Continual World. Our work is foundational research and does not lead to any direct negative applications.

## Acknowledgments and Disclosure of Funding

We would like to thank Stanisław Jastrzębski for stimulating talks and help while preparing the manuscript. The work of PM was supported by the Polish National Science Center grant UMO-2017/26/E/ST6/00622. The work of MW was funded by Foundation for Polish Science (grant no POIR.04.04.00-00-14DE/18-00 carried out within the Team-Net program co-financed by the European Union under the European Regional Development Fund. This research was supported by the PL-Grid Infrastructure. Our experiments were managed using `https://neptune.ai`. We would like to thank the Neptune team for providing us access to the team version and technical support.

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
