# A Technical details

## A.1 Tasks details

Continual World consists of a series of robotic tasks performed by a simulated Sawyer robot, each lasting for 200 steps. Individual tasks come from Meta-World [54] (benchmark released on MIT license) and as such, they have a significantly shared structure. This includes a common definition of the observation and action space as well as a similar reward structure.

**Observations**   The observation space is 12-dimensional, consisting of:

- the 3D Cartesian position of the robot end-effector (3 numbers),
- the 3D Cartesian positions of one or two objects that the robot manipulates[7] (6 numbers),
- the 3D Cartesian position of the goal to be reached (3 numbers).

**Actions**   The action space is 4-dimensional, including the requested 3D end-effector position change and the gripper actuator delta.

**Rewards**   Meta-World proposes a careful design of rewards, which ensures that tasks are solvable and exhibit shared structure across the tasks. These rewards are task-specific, see [54, Section 4.2, Table 3] and are used only in training.

In the evaluation phase, a binary success metric is used

$$\mathbb{I}_{\|o-g\|_2} < \varepsilon, \tag{5}$$

where $o, g$ are the 3D positions the object and goal, respectively, and $\varepsilon$ is a task-specific threshold, see [54, Table 4].

## A.2 Sequences

Our benchmark consists of two long CW10, CW20 sequences and 8 triples.

The CW10 sequence is: HAMMER-V1, PUSH-WALL-V1, FAUCET-CLOSE-V1, PUSH-BACK-V1, STICK-PULL-V1, HANDLE-PRESS-SIDE-V1, PUSH-V1, SHELF-PLACE-V1, WINDOW-CLOSE-V1, PEG-UNPLUG-SIDE-V1.

The CW20 is CW10 repeated twice. The triples are presented in Section F.

## A.3 Training and evaluation details

We use an implementation of SAC algorithm [17, 18] based on [1] , which we ported to TensorFlow 2. SAC is an actor-critic type algorithm that uses neural networks in two different roles: the policy network (actor) and the value function network (critic).

Both the actor and critic are randomly initialized at the start of the first task and then are trained through the whole sequence, retaining parameters from previous tasks.

In all cases, except for A-GEM and Perfect Memory, we use CL algorithms only for the policy networks, for details see Section C.1.

**Task handling**   Each task is trained for 1M steps. The replay buffer is emptied when a new task starts. For the first 10K steps, actions are sampled uniformly from the action space. After these, actions are sampled using the current policy. We use a warm-up period of 1K steps before training neural networks.

**SAC training**   We perform 50 optimization steps per each 50 interactions collected. To this end, we use the Adam optimizer [26]. Its internal statistics are reset for each task. The maximum entropy coefficient $\alpha$ is tuned automatically [18], so that the average standard deviation of Gaussian policy matches the target value $\sigma_t = 0.089$.

---

[7]For tasks with one object, the excessive coordinates are zeroed out.

| parameter | search space | selected value |
|---|---|---|
| learning rate | $\{3 \times 10^{-5}, 1 \times 10^{-4}, 3 \times 10^{-4}, 1 \times 10^{-3}\}$ | $1 \times 10^{-3}$ |
| batch size | $\{128, 256, 512\}$ | 128 |
| discount factor $\gamma$ | $\{0.95, 0.99, 0.995\}$ | 0.99 |
| target output std $\sigma_t$ | $\{0.03, 0.089, 0.3\}$ | 0.089 |
| replay buffer size | — | 1M |

Table 2: Search space for the common hyperparameters.

**Policy evaluation** Performance on all tasks is measured (mean of (5) on 10 stochastic trajectories) every $20K$ steps. In the case of some metrics, we average over 5 points in time to get smoother results.

## A.4 Network architecture

We use a network with 4 linear layers, each consisting of 256 neurons. We add Layer Normalization [7] after the first layer. We use leaky ReLU activations (with $\alpha = 0.2$) after every layer except the first one, where we use the tanh activation, which is known to work well with Layer Normalization layers.

The network has as many heads as the corresponding sequence, e.g. 20 for CW20. In Appendix G.3 we present experiments in which a single head is used, and the task id input is provided.

The presented architecture is used for both the actor and the critic. The actor network input consists of an observation, and it outputs the mean and log standard deviation of the Gaussian distribution on the action space. These outputs are produced for each task by its respective head. The critic network input consists of an observation and action, and it outputs a scalar value for each task head.

## A.5 Hyperparameters

We performed hyperparameter search in two stages. In the first stage, we selected hyperparameters of the SAC algorithm, which are common among all the methods. We measured the performance training on the CW20 sequence using only the fine-tuning method. As the metric, we used the mean performance at the end of the training of each task. We present the search space and selected parameters in Table 2.

In the second stage, we tuned method-specific hyperparameters using the final average performance as the objective. These are presented in Appendix B.

## A.6 Bootstrap confidence intervals

We use non-parametric bootstrap [15] to calculate confidence intervals. Reinforcement learning typically has rather high variability. We found that using 20 seeds usually results in informative intervals.

Unless specified otherwise, in all numerical results below, we use at least 20 seeds and report 90% confidence intervals.

## A.7 Infrastructure used

The typical configuration of a computational node used in our experiments was: the Intel Xeon E5-2697 2.60GHz processor with 128GB memory. On a single node, we ran 3 or 4 experiments at the same time. A single experiment on the CW20 sequence takes about 100 hours (with substantial variability with respect to the method used). We did not use GPUs; we found that with the relatively small size of the network, see Section A.4 it offers only slight wall-time improvement while generating substantial additional costs.

# B CL methods

We now describe seven CL methods evaluated on our benchmark: L2, EWC, MAS, VCL, PackNet, Perfect Memory and Reservoir Sampling. Most of them were developed in the supervised learning context and in many cases non-trivial adaptation to RL setting was required, which we describe in Appendix C. We picked the following approaches in an attempt to cover different families of methods considered in the community. Namely, following [13], we consider three classes: regularization-based methods, parameter isolation methods, and replay methods.

## B.1 Regularization-based methods

Regularization-based methods build on an observation that connectionist models, like neural networks, are heavily overparametized. A relatively small number of parameters is genuinely relevant for the previous tasks. Thus, if "protected", forgetting might be mitigated. Regularization methods implement this by an additional regularization loss, often in the $l_2$ form. For example, in order to remember task 1 while learning on task 2 we add a regularization cost of the form:

$$\lambda \sum_k F_k \left( \theta_k - \theta_k^1 \right)^2 , \tag{6}$$

where $\theta_k$ are the neural network current weights, $\theta_k^1$ are the values of weights after learning the first task, and $\lambda > 0$ is the overall regularization strength. The coefficients $F_k$ are critical, as they specify how important parameters are. Regularization-based methods are often derived from a Bayesian perspective, where the regularization term enforces the learning process to stay close to a prior, which now incorporates the knowledge acquired in the previous tasks.

There are multiple ways to extend this approach to multiple tasks. Arguably, the simplest one is to keep the importance vector $F^i$ and parameters $\theta^i$ for each task $i$. This, however, suffers from an unacceptable increase in memory consumption. We thus opt for a memory-efficiency solution presented in [22]. Namely, we store the sum of importance vectors and the weights after training the last task. Formally, when learning $m$-th task, the penalty has the following form:

$$\lambda \sum_k \left( \sum_{i=1}^{m-1} F_k^i \right) \left( \theta_k - \theta_k^{m-1} \right)^2 . \tag{7}$$

**L2** This method is used as a baseline in [27]. It declares all parameters to be equally important, i.e. $F_k^i = 1$ for all $k, i$. Despite its simplicity, it is able to reduce forgetting, however, often at the cost of a substantial reduction in the ability to learn.

For the L2 method, we tested the following hyperparameter values: $\lambda \in \{10^{-2}, 10^{-1}, 10^0, 10^1, 10^2, 10^3, 10^4, 10^5\}$; selected value is $10^5$.

**EWC** Elastic Weight Consolidation [27] adheres to the Bayesian perspective. Specifically, it proposes that the probability distribution of parameters of the previous tasks is a prior when learning the next task. Since this distribution is intractable, it is approximated using the diagonal of the Fisher Information Matrix. Namely, $F_k = \mathbb{E}_{x \sim \mathcal{D}} \mathbb{E}_{y \sim p_\theta(\cdot|x)} \left( \nabla_{\theta_k} \log p_{\theta_k}(y|x) \right)^2$. The outer expectation is approximated with a sample of 2560 examples from the replay buffer $\mathcal{D}$. The inner expectation can be calculated analytically for Gaussian distributions used in this work, see Section C.2 for details. We clipped the calculated values $F_k$ from below so that the minimal value is $10^{-5}$.

For the EWC method we tested the following hyperparameter values: $\lambda \in \{10^{-2}, 10^{-1}, 10^0, 10^1, 10^2, 10^3, 10^4, 10^5\}$; selected value is $10^4$.

**MAS** Memory Aware Synapses [4] estimates the importance of each neural network weight by measuring the sensitivity of the output with respect to the weight perturbations. Formally, $F_k = \mathbb{E}_{x \sim \mathcal{D}} \left( \frac{\partial [\|g(x)\|_2^2]}{\partial \theta_k} \right)$, where $g$ is the output of the model and expectation is with respect to the data distribution $\mathcal{D}$. To approximate the expected value in the formula above, we sampled 2560 examples from the buffer.

For the MAS method we tested the following hyperparameter values: $\lambda \in \{10^{-2}, 10^{-1}, 10^0, 10^1, 10^2, 10^3, 10^4, 10^5\}$; selected value is $10^4$.

**VCL** Variational Continual Learning [37] builds on the Bayesian neural network framework. Specifically, it maintains a factorized Gaussian distribution over the network parameters and applies the variational inference to approximate the Bayes rule update. Thus, the training objective contains an additional loss component

$$\lambda D_{\mathrm{KL}}(\theta \parallel \theta^{m-1}), \tag{8}$$

where $D_{\mathrm{KL}}$ denotes the Kullback-Leibler divergence and $\theta^{m-1}$ is the parameter distribution corresponding to the previous tasks. This loss can be viewed as regularization term similar to (6).

We introduced multiple changes to make VCL usable in the RL setting. First, we tune $\lambda$. The original work sets $\lambda = 1/N$, where $N$ is the number of samples in the current task, which we found performing poorly. Second, we discovered that not using prior (equivalently, setting $\lambda = 0$) on the first task yields much better overall results. Third, along with a standard smoothing procedure, [9] averages the prediction using 10 samples of the weights. We opted for taking 1 sample, since it performed comparably with 10 samples, but significantly decreased the training time. Fourth, we do not use the coresets mechanism due to a high computational cost. Further discussion of these issues is presented in Section C.3.

For the VCL method we initialize parameters with $\mathcal{N}(\mu, \sigma)$, $\mu = 0, \sigma = 0.025$. We tested the following hyperparameter values: $\lambda \in \{10^{-7}, 10^{-6}, 10^{-5}, 10^{-4}, 10^{-3}, 10^{-2}, 10^{-1}, 1\}$; selected value is $1$.

### B.2 Parameter isolation methods

Parameter isolation methods (also called modularity-based methods) retain already acquired knowledge by keeping certain ranges of parameters fixed. Such an approach may be considered as a hard constraint, as opposed to the soft penalties of regularization-based methods.

**PackNet** Introduced in [33], this method iteratively applies pruning techniques after each task is trained. In this way, it "packs the task" into a subset of the neural network parameters leaving others available for the subsequent tasks; the parameters associated with the previous tasks are frozen. Note that by design, this method completely avoids forgetting. Compared to earlier works, such as progressive networks [43], the model size stays fixed through learning. However, with every task, the number of available parameters shrinks.

Pruning is a two-stage process. In the first one, a fixed-sized subset of parameters most important for the task is chosen. The size is set to $5\%$ in the case of CW20. In the second stage, the network spanned by this subset is fine-tuned for a certain number of steps. In this process, we only use the already collected data.

All biases and normalization parameters (of layer normalization) are not handled by PackNet. They remain frozen after training the first task. Also, the last layer is not managed by PackNet, as each task has its own head.

In PackNet we used global gradient norm clipping $2 \times 10^{-5}$, as explained in Section C.6.

For PackNet, we tested the following hyperparameter values: number of fine-tuning steps in $\{50K, 100K, 200K\}$; selected value if $100K$.

### B.3 Replay methods

Replay methods maintain a buffer of samples from the previous tasks and use them to reduce forgetting. These are somewhat similar but should not be confused with the replay buffer technique commonly used by off-policy RL algorithms (including SAC).

**Reservoir Sampling** Reservoir sampling [53] is an algorithm, which manages the buffer so that its distribution approximates the empirical distribution of observed samples. This method is suitable for CL and often used as a standard mechanism for gathering samples [12]. In our implementation, we augment samples with the task id and store them in the SAC replay buffer. For the sake of simplicity, in the main part of the paper we stick to an otherwise unrealistic assumption of the unlimited capacity buffer, which we dubbed as **Perfect Memory**.

We find that it does not work very well, even if we increase the number of samples replayed at each step (batch size) to 512, which significantly extends the training time.

For the reservoir sampling methods, we tested the following hyperparameter values: batch size $\in \{64, 128, 256, 512\}$ (selected value $= 512$), replay size $\in \{5 \cdot 10^6, 1 \cdot 10^7, 2 \cdot 10^7\}$ (selected value $= 2 \cdot 10^7$). The value $2 \cdot 10^7$ corresponds to Perfect Memory, as it is able to store all experience of CW20.

**A-GEM**  Averaged Gradient Episodic Memory [12] frames continual learning as a constrained optimization problem. Formally, a loss for the current task $\ell(\theta, \mathcal{D})$ is to be minimized under the condition that the losses on the previous tasks are bounded $\ell(\theta, \mathcal{M}_k) \leq \ell_k$, where $\ell_k$ is the previously observed minimum and $\mathcal{M}_k$ are the buffer samples for task $k$. Unfortunately, such constraint is not tractable for neural networks. [12, Section 4] proposes an approximation based on the first-order Taylor approximation:

$$\langle \nabla_\theta \ell(\theta, \mathcal{B}_{new}), \nabla_\theta \ell(\theta, \mathcal{B}_{old}) \rangle > 0, \tag{9}$$

where $\mathcal{B}_{new}, \mathcal{B}_{old}$ are, respectively, batches of data for the current and previous tasks. This constraint can be easily implemented by a gradient projection:

$$\nabla_\theta \ell(\theta, \mathcal{B}_{new}) - \frac{\langle \nabla_\theta \ell(\theta, \mathcal{B}_{new}), \nabla_\theta \ell(\theta, \mathcal{B}_{old}) \rangle}{\langle \nabla_\theta \ell(\theta, \mathcal{B}_{old}), \nabla_\theta \ell(\theta, \mathcal{B}_{old}) \rangle} \nabla_\theta \ell(\theta, \mathcal{B}_{old}) \tag{10}$$

For A-GEM, we set episodic memory per task to 10K. We tested the following hyperparameter values: episodic batch size $\in \{128, 256\}$ (selected value $= 128$).

## B.4  Resources comparison

As we highlight in Section 3, continual learning is characterized by multiple, often conflicting, desiderata. In practice, except for the metrics studied in the paper (see Section 4.1), one could pay attention to the efficiency in terms of resource usage. To this end, we check how the tested methods perform in terms of computational power, measured by average wall time and memory efficiency, measured by the overhead in the number of parameters, and the number of examples in the buffer. The results presented in Table 3 show that most of the methods use a fairly modest amount of resources, exceptions being reservoir sampling, which needs twice as much computation time as the baseline fine-tuning, and VCL, which requires more parameters due to using Bayesian neural networks.

| Method | Normalized wall time | Normalized parameter overhead | Examples to remember |
|---|---|---|---|
| Fine-tuning | 1.00 | 1.0 | N/A |
| Reservoir | 2.20 | 1.0 | 20M |
| A-GEM | 1.47 | 1.0 | 1M |
| EWC | 1.13 | 2.0 (parameters of the previous task & importance weights) | N/A |
| L2 | 1.12 | 1.5 (parameters of the previous task) | N/A |
| MAS | 1.13 | 2.0 (parameters of the previous task & importance weights) | N/A |
| PackNet | 1.15 | 1.5(*) (mask assigning parameters to task) | N/A |
| VCL | 1.28 | 4.0 (parameters of the previous task & network is two times bigger as we model standard deviation) | N/A |

Table 3: Results for computational and memory overhead for each method (normalized with respect to Fine-tuning). (*)Masks in PackNet (integers) can be encoded with fewer bits than the parameters (floats).

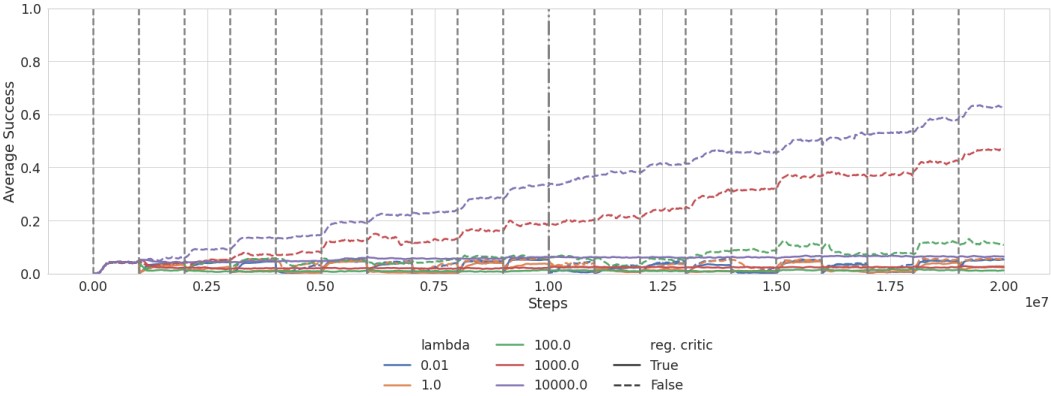

Figure 5: Average performance for EWC with various settings of the critic regularization, see (6). The reference dashed lines show the performance without critic regularization.

# C    Algorithmic and conceptual challenges of the RL+CL setting

This section lists various challenges that we encountered while implementing the CL methods described in Section B. In most cases, they stem from the fact that the methods were initially developed for supervised learning and needed to be adopted to the more algorithmically complex RL setting.

## C.1    Critic regularization

One of perhaps most important algorithmic conclusions is that *regularizing the critic network is harmful to performance*. We recall that in all methods, except for AGEM and Perfect Memory, the critic is only used during the training. Consequently, it needs to store only the value function for the current task, and thus forgetting is not a (primary) issue. Further, one can opt for not using any CL method on the critic network. The rationale for such a decision is to maintain the full network "plasticity" (which is typically reduced by CL methods). In our experiments, we show that this is the case. However, we stress that not regularizing critic might be controversial. A suggestive straightforward example, present in the CW20 sequence, is a situation when a task is repeated after a while, and the critic needs to be trained from scratch.

We now present an in-depth analysis using EWC as an example. To this end, we performed a separate hyperparameter search for this setting for the regularization parameter $\lambda$ in (6). Results, as presented in Figure 5, clearly indicate that the regularized cases struggle to achieve a decent result, in stark contrast to the non-regularized references (dashed lines) (we tested more values of $\lambda$, which are not included for better readability).

We consider critic regularization issues to be an important research direction.

## C.2    EWC Fisher matrix derivation

For the sake of completeness, we provide an analytical derivation of the Fisher matrix diagonal coefficients in the case of Gaussian distribution used by the SAC policy. In experiments presented in Section C.1 we use these formulas also for the critic network assuming a normal distribution with constant $\sigma^2 = 1$.

**Fact 1.** *Let* $\mu : \mathbb{R} \mapsto \mathbb{R}, \sigma : \mathbb{R} \mapsto \mathbb{R}$ *parameterize a Gaussian distribution* $\theta \mapsto \mathcal{N}\left(\mu(\theta), \sigma^2(\theta)\right)$. *Then the diagonal of the Fisher information matrix* $\mathcal{I}$ *is*

$$\mathcal{I}_{kk} = \left(\frac{\partial \mu}{\partial \theta_k} \cdot \frac{1}{\sigma}\right)^2 + 2\left(\frac{\partial \sigma}{\partial \theta_k} \cdot \frac{1}{\sigma}\right)^2. \tag{11}$$

Note that in SAC we use a factorized multivariate Gaussian distribution with mean vector $(\mu_1, \ldots, \mu_N)$ and standard deviations $(\sigma_1, \ldots, \sigma_N)$. It is straightforward to check that this case

$$\mathcal{I}_{kk} = \sum_{\ell=1}^{N} \left( \frac{\partial \mu_\ell}{\partial \theta_k} \cdot \frac{1}{\sigma_\ell} \right)^2 + 2 \left( \frac{\partial \sigma_\ell}{\partial \theta_k} \cdot \frac{1}{\sigma_\ell} \right)^2.$$

*Proof.* We fix parameters $\theta_0$, denote $\mu_0 = \mu(\theta_0), \sigma_0 = \sigma(\theta_0)$ and $\mathcal{N}_0 := \mathcal{N}_0(\mu_0, \sigma_0^2)$. We will calculate the the diagonal Fisher matrix coefficient at $\theta_0$ using the fact that it is the curvature of the Kullback–Leibler divergence:

$$\mathcal{I}_{kk} = \frac{\partial^2}{\partial \theta_k^2} F(\theta), \quad F(\theta) := f(\mu(\theta_i), \sigma(\theta_i)), \tag{12}$$

where

$$f(\mu, \sigma) := D_{\text{KL}}\big(\mathcal{N}_0 \parallel \mathcal{N}(\mu, \sigma^2)\big) = \frac{1}{2} \left\{ \left( \frac{\sigma_0}{\sigma} \right)^2 + \frac{(\mu - \mu_0)^2}{\sigma^2} - 1 + 2 \ln \frac{\sigma}{\sigma_0} \right\}.$$

We start with

$$\frac{\partial f}{\partial \mu} = \frac{(\mu - \mu_0)}{\sigma^2}, \quad \text{thus} \quad \frac{\partial f}{\partial \mu}(\mu_0, \sigma_0) = 0. \tag{13}$$

Further

$$\frac{\partial^2 f}{\partial \mu^2} = \frac{1}{\sigma^2}, \quad \text{thus} \quad \frac{\partial f}{\partial \mu}(\mu_0, \sigma_0) = \frac{1}{\sigma_0^2}. \tag{14}$$

For $\sigma$ we have

$$\frac{\partial f}{\partial \sigma} = \frac{1}{2} \left\{ -2\frac{\sigma_0^2}{\sigma^3} - 2\frac{(\mu - \mu_0)^2}{\sigma^3} + \frac{2}{\sigma} \right\} = -\frac{\sigma_0^2}{\sigma^3} - \frac{(\mu - \mu_0)^2}{\sigma^3} + \frac{1}{\sigma}, \tag{15}$$

$$\frac{\partial f}{\partial \sigma}(\mu_0, \sigma_0) = -\frac{\sigma_0^2}{\sigma_0^3} - \frac{0}{\sigma^3} + \frac{1}{\sigma_0} = 0.$$

Further

$$\frac{\partial^2 f}{\partial \sigma^2} = 3\frac{\sigma_0^2}{\sigma^4} + 3\frac{(\mu - \mu_0)^2}{\sigma^4} - \frac{1}{\sigma^2}, \quad \text{thus} \quad \frac{\partial^2 f}{\partial \sigma^2}(\mu_0, \sigma_0) = 3\frac{\sigma_0^2}{\sigma_0^4} + 3\frac{0}{\sigma^4} - \frac{1}{\sigma_0^2} = \frac{2}{\sigma_0^2}. \tag{16}$$

Now, we come back to (12). Applying the chain rule

$$\frac{\partial^2 F}{\partial \theta_k^2} = \frac{\partial^2 \mu}{\partial \theta_k^2} \cdot \frac{\partial f}{\partial \mu} + \left( \frac{\partial \mu}{\partial \theta_k} \right)^2 \cdot \frac{\partial^2 f}{\partial \mu^2} + \frac{\partial^2 \sigma}{\partial \theta_k^2} \cdot \frac{\partial f}{\partial \sigma} + \left( \frac{\partial \sigma}{\partial \theta_k} \right) \cdot \frac{\partial^2 f}{\partial \sigma^2}.$$

When evaluating at $\theta = \theta_0$ the first and third terms vanish, by (13) and (15). Thus

$$\frac{\partial^2 F}{\partial \theta_k^2}(\theta_0) = \left( \frac{\partial \mu}{\partial \theta_k}(\theta_0) \right)^2 \cdot \frac{\partial^2 f}{\partial \mu^2}(\theta_0) + \left( \frac{\partial \sigma}{\partial \theta_k}(\theta_0) \right)^2 \cdot \frac{\partial^2 f}{\partial \sigma^2}(\theta_0).$$

Now we use (15) and (16) we arrive at (11).

$\square$

## C.3 VCL

We suggest introducing two changes to the original VCL setup [37]: treating $\lambda$ as hyperparameter and not using prior when training the first task. Our conclusion follows from the following three experimental evaluations:

1. Fixed $\lambda = 1/N$, where $N = 10^6$ is the number of samples in each task. The prior is applied during the first task.

2. We tested the following hyperparameter values: $\lambda \in \{10^{-7}, 10^{-6}, 10^{-5}, 10^{-4}, 10^{-3}, 10^{-2}, 10^{-1}, 10^0\}$; selected value is $10^{-5}$. The prior is applied during the first task.

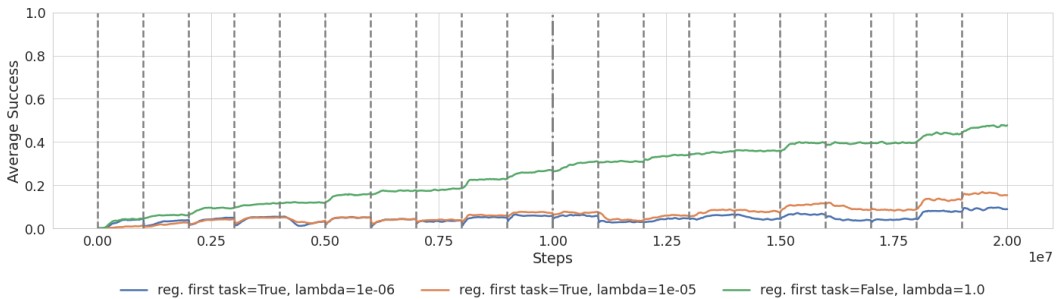

Figure 6: Average performance for VCL.

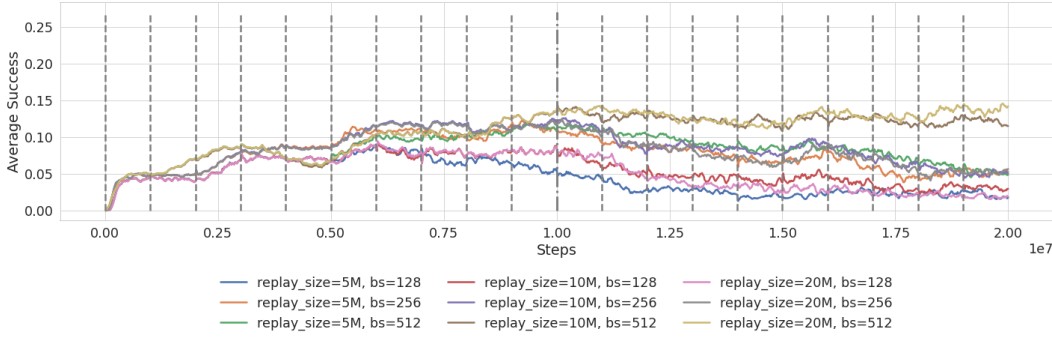

Figure 7: The average performance of Reservoir Sampling. Note the $y$-axis scale change; $bs$ denotes the batch size.

3. We tested the following hyperparameter values: $\lambda \in \{10^{-7}, 10^{-6}, 10^{-5}, 10^{-4}, 10^{-3}, 10^{-2}, 10^{-1}, 10^{0}\}$; selected value is 1. The prior is *not* applied during the first task.

The first setup is taken from [37]. As can be seen in Figure 6 it underperformed, failing to keep good performance on previous tasks. Increasing $\lambda$ to $10^{-5}$ improves the results slightly. We observed that higher values of $\lambda$ fail due to strong regularization with respect to the initial prior. Abandoning this prior has led us to the most successful setup. Interestingly, the much higher value of $\lambda = 1.0$ is optimal here.

## C.4 Reservoir Sampling

We present more experiments for the Reservoir Sampling method. We vary the size of the replay $\in \{5M, 10M, 20M\}$. Recall that 20M is enough to store the whole experience (which corresponds to the Perfect Memory experiments in the main section). For smaller sizes, we use reservoir sampling as described in Section B.3. We also test for different batch size values $\{64, 128, 256, 512\}$. We observe that higher values result in better performance, possibly due to better data balance in a single batch. This, however, comes at the cost of training speed.

Figure 7 shows results for different hyperparameter settings. Runs with the standard batch size 128 used by the rest of the methods fail to exceed 0.05 final average success obtained by Fine-tuning. Increasing the batch size improves the results significantly, and the two best runs using batch size 512 achieve final average performance between 0.1 and 0.15. However, we note that even the best performing hyperparameters for Reservoir Sampling do not achieve as good results as the regularization and parameter isolation methods. We attribute this phenomenon to the critic regularization problem discussed in Section C.1.

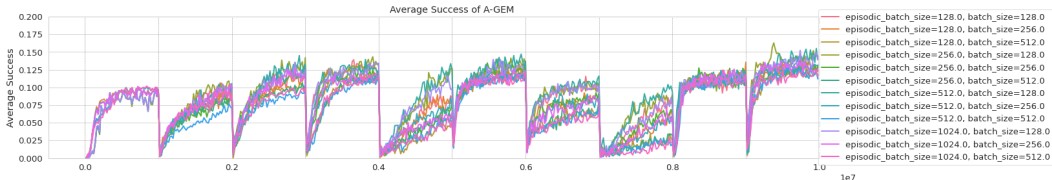

Figure 8: Average performance of A-GEM on CW10. Note the $y$-axis scale change.

## C.5  A-GEM not working

Despite our best efforts, A-GEM has poor performance on our benchmark. In order to better investigate this case, we conduct additional analysis. In Figure 8 we show average performance on CW10 for various hyperparameter settings. We set the size of episodic task memory to 100k samples per task. We search for batch size (number of samples used to compute $\ell(\theta, \mathcal{B}_{new})$) in $\{128, 256, 512\}$ and for the episodic batch size (number of samples used to compute $\ell(\theta, \mathcal{B}_{old})$) in $\{128, 256, 512, 1024\}$, see (10). The results show that none of those settings perform significantly better than the others. Similar to Reservoir Sampling, we attribute those problems to the critic regularization discussed in Section C.1. Another conjecture is that due to the high variance of RL gradients (higher than in the supervfised learning setting), approximation (9) is too brittle to work well.

## C.6  PackNet extensive clipping

Initially, PackNet was unstable, degenerating (NaNs and infinities) near the task boundaries. We amended this problem by applying global gradient norm clipping $2 \times 10^{-5}$. We tune this value so that the learning does not explode and yet the performance is mostly preserved. PackNet instability requires further investigation: we conjecture that frequent changes of the set of optimized parameters might amplify the inherent instability of temporal-difference learning.

# D  Transfer matrix

In Table 4 we show the transfer matrix corresponding to Figure 2 (Section 4.2). Recall that the transfer matrix is aimed to measure the relationship between each two tasks (for transfer on CW20 see Table 8). Most entries in the Table 4 are positive, however, some tasks seem to interfere with one another (most notably PEG-UNPLUG-V1 (9) followed by PUSH-V1 (6)). Some of the factors influencing the transfer are: similarity in visual representation (e.g. objects on the scene), reward structure (reach, grasp, place), or direction in which the robotic arm moves. The average forward transfer given the second task (columns) is more variable than the corresponding quantity for the first task (rows), with the standard deviation equal to 0.21 and 0.07, respectively.

## D.1  Diagonal transfers

The transfers on the diagonal (i.e., between the same tasks) are smaller than expected. We enumerate possible reasons: using multi-head architecture, buffer resetting, and uniform sampling exploration policy for the second task. These design elements are generally required when switching tasks, but not on the diagonal, where we resume the same task. This makes the diagonal of the transfer matrix suitable for an in-depth ablation analysis.

The experiments for the whole diagonal are summarized in Table 5 and example curves are shown in Figure 9. We tested four settings. Multi Head (reset buffer) is our standard setting. Changing to a single head network, dubbed as Single Head (reset buffer), brings no significant difference. However, keeping the examples in the buffer in Single Head (no buffer reset) yields a major improvement. Similarly, in Single Head (no random exploration), when we do not perform random exploration at the task change the performance improves. We conjecture, that resetting the buffer and performing random exploration results in distributional shift, which is hard to be handled by SAC.

| Second task / First task | 0 | 1 | 2 | 3 | 4 | 5 | 6 | 7 | 8 | 9 | mean |
|---|---|---|---|---|---|---|---|---|---|---|---|
| 0 | 0.42 [0.32, 0.51] | 0.13 [-0.01, 0.24] | 0.64 [0.56, 0.70] | 0.58 [0.35, 0.76] | 0.24 [0.12, 0.35] | 0.48 [0.39, 0.56] | 0.03 [-0.09, 0.14] | 0.58 [0.54, 0.62] | 0.32 [0.26, 0.38] | 0.47 [0.05, 0.81] | 0.39 |
| 1 | 0.13 [0.01, 0.25] | -0.05 [-0.18, 0.05] | 0.32 [0.11, 0.48] | 0.76 [0.72, 0.79] | 0.01 [-0.09, 0.11] | 0.46 [0.38, 0.53] | -0.04 [-0.12, 0.04] | 0.28 [0.21, 0.35] | 0.36 [0.30, 0.42] | 0.78 [0.69, 0.85] | 0.30 |
| 2 | 0.04 [-0.08, 0.15] | -0.04 [-0.10, 0.03] | 0.50 [0.35, 0.63] | 0.75 [0.72, 0.78] | 0.01 [-0.08, 0.10] | 0.26 [0.19, 0.33] | -0.17 [-0.30, 0.06] | 0.32 [0.22, 0.41] | 0.27 [0.17, 0.35] | 0.79 [0.71, 0.85] | 0.27 |
| 3 | 0.03 [-0.22, 0.24] | -0.15 [-0.30, 0.01] | 0.40 [0.14, 0.61] | 0.47 [0.23, 0.67] | 0.09 [-0.00, 0.18] | 0.42 [0.34, 0.49] | 0.01 [-0.10, 0.11] | 0.39 [0.29, 0.48] | 0.32 [0.24, 0.40] | 0.33 [-0.08, 0.66] | 0.23 |
| 4 | 0.19 [-0.02, 0.36] | 0.21 [0.14, 0.28] | 0.45 [0.20, 0.63] | 0.47 [0.22, 0.70] | 0.42 [0.34, 0.50] | 0.41 [0.32, 0.50] | 0.19 [0.11, 0.26] | 0.53 [0.46, 0.59] | 0.36 [0.28, 0.43] | 0.38 [-0.05, 0.71] | 0.36 |
| 5 | 0.05 [-0.08, 0.16] | -0.00 [-0.14, 0.11] | 0.55 [0.43, 0.65] | 0.64 [0.49, 0.74] | 0.05 [-0.04, 0.13] | 0.51 [0.45, 0.56] | -0.14 [-0.25, -0.05] | 0.20 [0.06, 0.33] | 0.32 [0.25, 0.38] | -0.01 [-0.61, 0.50] | 0.22 |
| 6 | 0.20 [0.08, 0.31] | -0.06 [-0.16, 0.03] | 0.47 [0.35, 0.57] | 0.78 [0.75, 0.81] | 0.09 [-0.03, 0.19] | 0.55 [0.50, 0.60] | -0.01 [-0.08, 0.06] | 0.30 [0.18, 0.41] | 0.42 [0.35, 0.48] | 0.81 [0.77, 0.85] | 0.36 |
| 7 | 0.29 [0.09, 0.44] | -0.03 [-0.14, 0.07] | 0.33 [0.07, 0.53] | 0.20 [-0.10, 0.47] | 0.14 [0.04, 0.24] | 0.42 [0.35, 0.48] | -0.06 [-0.17, 0.05] | 0.40 [0.31, 0.49] | 0.33 [0.24, 0.41] | 0.10 [-0.48, 0.57] | 0.21 |
| 8 | -0.10 [-0.33, 0.10] | 0.06 [0.00, 0.12] | 0.24 [0.04, 0.41] | 0.69 [0.63, 0.74] | -0.01 [-0.11, 0.08] | 0.22 [0.15, 0.28] | -0.02 [-0.08, 0.04] | 0.42 [0.34, 0.49] | 0.36 [0.29, 0.43] | 0.80 [0.75, 0.84] | 0.27 |
| 9 | 0.10 [-0.09, 0.26] | -0.05 [-0.15, 0.04] | 0.30 [0.06, 0.48] | 0.53 [0.34, 0.68] | -0.02 [-0.11, 0.08] | 0.18 [0.08, 0.26] | -0.28 [-0.40, -0.16] | 0.21 [0.12, 0.30] | 0.31 [0.25, 0.37] | 0.74 [0.63, 0.83] | 0.20 |
| mean | 0.14 | 0.00 | 0.42 | 0.59 | 0.10 | 0.39 | -0.05 | 0.36 | 0.34 | 0.52 | 0.28 |

Table 4: Transfer matrix. Normalized forward transfers for the second task. For task names see Section A.2.

# E  CW20 results

In Table 6, we recall for convenience the summary of our results for the experiments on the CW20 sequence (additional backward transfer results are described in Section E.1). We also recall that the methods were tuned with the objective of maximizing the final average performance. Our main findings are that most methods are efficient with mitigating forgetting but have an unsatisfying forward transfer. The best method, PackNet, has forward transfer close to Fine-tuning, which is a strong baseline for this metric. However, both fall below the reference value obtained from the analysis of the transfer matrix RT = 0.46.

Table 6 additionally highlights an important fact that CL metrics are interrelated. For instance, high forward transfer and low performance (a case for Fine-tuning and A-GEM) have to imply high forgetting.

| Task | MH (reset buffer) | SH (reset buffer) | SH (no buffer reset) | SH (no random exp) |
|---|---|---|---|---|
| `hammer-v1` | 0.42 | 0.26 | 0.56 | 0.74 |
| `push-wall-v1` | -0.05 | 0.06 | 0.55 | 0.40 |
| `faucet-close-v1` | 0.50 | 0.64 | 0.60 | 0.78 |
| `push-back-v1` | 0.47 | 0.41 | 0.67 | 0.68 |
| `stick-pull-v1` | 0.42 | 0.37 | 0.60 | 0.63 |
| `handle-press-side-v1` | 0.51 | 0.53 | 0.53 | 0.68 |
| `push-v1` | -0.01 | -0.03 | 0.64 | 0.35 |
| `shelf-place-v1` | 0.40 | 0.23 | 0.29 | 0.40 |
| `window-close-v1` | 0.36 | 0.50 | 0.45 | 0.52 |
| `peg-unplug-side-v1` | 0.74 | 0.83 | 0.97 | 0.96 |
| mean | 0.38 | 0.38 | 0.58 | 0.61 |

Table 5: Transfers on the diagonal. MH and SH stand for multi-head and single-head, respectively.

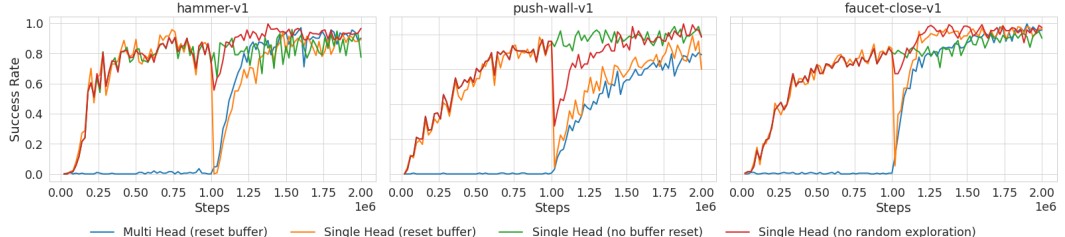

Figure 9: Success rates for example tasks on the diagonal of the transfer matrix. The blue line is the standard setup. Note, that its poor performance in the first part of each plot is expected, as we evaluate the head which has not yet been trained. Performance improves when we keep the samples in the buffer when switching the task or when we disable the exploration through uniform policy.

| method | performance | forgetting | f. transfer | b. transfer |
|---|---|---|---|---|
| **Fine-tuning** | 0.05 [0.05, 0.06] | 0.73 [0.72, 0.75] | **0.20 [0.17, 0.23]** | 0.00 [0.00, 0.00] |
| **L2** | 0.43 [0.39, 0.47] | 0.02 [0.00, 0.03] | -0.71 [-0.87, -0.57] | 0.02 [0.02, 0.03] |
| **EWC** | 0.60 [0.57, 0.64] | 0.02 [-0.00, 0.05] | -0.17 [-0.24, -0.11] | 0.04 [0.03, 0.04] |
| **MAS** | 0.51 [0.49, 0.53] | 0.00 [-0.01, 0.02] | -0.52 [-0.59, -0.47] | 0.04 [0.04, 0.05] |
| **VCL** | 0.48 [0.46, 0.50] | 0.01 [-0.01, 0.02] | -0.49 [-0.57, -0.42] | 0.04 [0.03, 0.04] |
| **PackNet** | **0.80 [0.79, 0.82]** | 0.00 [-0.01, 0.01] | **0.19 [0.15, 0.23]** | 0.03 [0.03, 0.04] |
| **Perfect Memory** | 0.12 [0.09, 0.15] | 0.07 [0.05, 0.10] | -1.34 [-1.42, -1.27] | 0.01 [0.01, 0.01] |
| **A-GEM** | 0.07 [0.06, 0.08] | 0.71 [0.70, 0.73] | 0.13 [0.10, 0.16] | 0.00 [0.00, 0.00] |
| **MT** | 0.51 [0.48, 0.53] | — | — | — |
| **MT (PopArt)** | 0.65 [0.63, 0.67] | — | — | — |
| **RT** | — | — | **0.46** | — |

Table 6: Results on CW20, for CL methods and multi-task training.

We recall that CW10 and CW20 sequences are defined in Section A.2. In the next sections, we discuss the results in more detail. We also provide the following visualizations:

- Figure 14 - performance curves averaged over tasks.
- Figure 15 - performance curves for the active task.
- Figure 16 - performance curves for all tasks. Useful for qualitative studies of transfer and forgetting.
- Figure 17 - visualization of forward transfer for each task.
- Figure 18 - forgetting curves averaged over tasks.

### E.1 Forgetting and backward transfer

Table 7 expands upon Table 6 by presenting more detailed forgetting results, see also Figure 18 and Figure 16. The latter is convenient to observe the evaluation dynamics of each task.

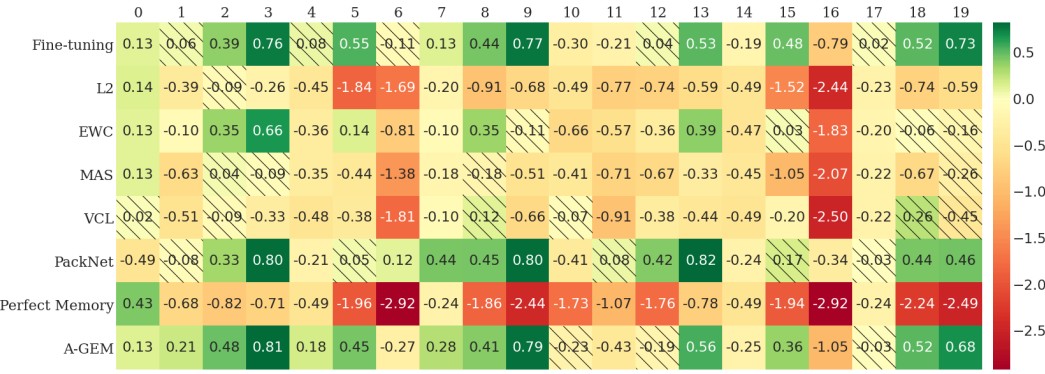

Figure 10: Forward transfer heatmap for CW20. The shaded cells indicate that $0$ belongs to the corresponding $90\%$ confidence interval.

Fine-tuning and AGEM exhibit rapid catastrophic forgetting after task switch (see Figure 16). It is expected for the former, but quite surprising for the latter. We conjecture possible reasons in Section C.5. Some mild forgetting can also be observed for Perfect Memory (it is not directly observed in the metric due to the poor training but can be visible on the graphs). The rest of the methods are quite efficient in mitigating forgetting. This also includes the very basic L2 method (which is at the cost of poor transfer, however).

In our experiment, we do not observe backward transfer, see Table 6. The values were calculated according to $B := \frac{1}{N} \sum_{i=1}^{N} B_i$, where

$$B_i := \max\left\{0, p_i(T) - p_i(i \cdot \Delta)\right\},$$

see Section 4.1 for notation. Note that some small values are possible due to stochastic evaluations.

## E.2  Forward transfer

Table 8 and Figure 10 supplement Table 6 with detailed results on forward transfer on CW20. Corresponding training curves can be find in Figure 17 and Figure 15.

Fine-tuning is a strong baseline, with PackNet and AGEM taking close second and third place (although the latter method has a low overall performance). Unfortunately, all the other methods suffer from the negative forward transfer, which in all cases, except possibly from EWC, is quite significant. To make the picture grimmer, transfer on the second part of CW20 is worse, even though the task has been learned (and not forgotten) on the first part, see Table 9.

## E.3  Results for CW10

In Table 10 we present results for the CW10 sequence (10 task version of the benchmark, without repeating the tasks). One can see that results are mostly consistent with CW20 (see Table 6), and thus CW10 can be used for faster experimenting. More detailed transfer results can be found in Table 8.

| task | Fine-tuning | L2 | EWC | MAS | VCL | PackNet | Perfect Memory | A-GEM |
|---|---|---|---|---|---|---|---|---|
| 0 | 0.86 [0.82, 0.90] | 0.07 [-0.04, 0.19] | 0.38 [0.23, 0.53] | -0.02 [-0.07, 0.04] | -0.05 [-0.10, 0.01] | -0.01 [-0.09, 0.06] | 0.56 [0.48, 0.64] | 0.86 [0.82, 0.91] |
| 1 | 0.72 [0.67, 0.77] | 0.08 [0.02, 0.14] | 0.04 [-0.04, 0.13] | 0.01 [-0.07, 0.08] | 0.04 [-0.02, 0.09] | -0.03 [-0.07, 0.00] | 0.29 [0.19, 0.40] | 0.78 [0.72, 0.83] |
| 2 | 0.91 [0.88, 0.95] | 0.09 [0.01, 0.18] | 0.01 [-0.05, 0.08] | 0.01 [-0.07, 0.09] | 0.01 [-0.05, 0.07] | -0.03 [-0.06, -0.01] | 0.07 [0.03, 0.12] | 0.92 [0.88, 0.96] |
| 3 | 0.99 [0.98, 0.99] | -0.00 [-0.02, 0.02] | 0.04 [0.01, 0.07] | 0.00 [-0.07, 0.08] | -0.04 [-0.08, -0.00] | -0.01 [-0.01, -0.00] | 0.04 [0.00, 0.12] | 0.98 [0.96, 0.99] |
| 4 | 0.67 [0.55, 0.77] | -0.01 [-0.03, 0.00] | -0.03 [-0.08, 0.02] | -0.10 [-0.17, -0.05] | -0.01 [-0.03, 0.01] | -0.03 [-0.09, 0.02] | 0.00 [0.00, 0.00] | 0.76 [0.67, 0.84] |
| 5 | 0.96 [0.95, 0.97] | -0.01 [-0.03, 0.00] | 0.01 [-0.06, 0.11] | -0.01 [-0.10, 0.07] | 0.13 [0.01, 0.25] | -0.02 [-0.03, 0.00] | 0.04 [-0.01, 0.08] | 0.97 [0.95, 0.99] |
| 6 | 0.83 [0.80, 0.86] | 0.02 [-0.02, 0.06] | 0.02 [-0.02, 0.06] | 0.04 [0.00, 0.08] | -0.00 [-0.05, 0.05] | 0.04 [-0.00, 0.09] | 0.00 [0.00, 0.00] | 0.78 [0.75, 0.81] |
| 7 | 0.54 [0.44, 0.64] | 0.01 [-0.02, 0.05] | 0.01 [-0.04, 0.06] | 0.01 [-0.02, 0.04] | -0.02 [-0.06, 0.02] | 0.06 [-0.00, 0.13] | 0.00 [0.00, 0.00] | 0.66 [0.49, 0.80] |
| 8 | 0.87 [0.83, 0.90] | -0.01 [-0.03, 0.02] | 0.02 [-0.07, 0.12] | 0.07 [-0.01, 0.17] | -0.06 [-0.12, 0.01] | -0.00 [-0.02, 0.01] | 0.01 [-0.07, 0.08] | 0.89 [0.84, 0.94] |
| 9 | 0.97 [0.96, 0.98] | 0.01 [-0.01, 0.04] | -0.00 [-0.01, 0.00] | 0.03 [-0.04, 0.11] | 0.11 [0.03, 0.20] | 0.00 [-0.00, 0.01] | 0.01 [-0.02, 0.04] | 0.93 [0.88, 0.98] |
| 10 | 0.67 [0.54, 0.80] | 0.05 [0.00, 0.12] | -0.08 [-0.12, -0.04] | -0.01 [-0.03, 0.01] | 0.05 [0.00, 0.12] | -0.02 [-0.06, 0.03] | 0.00 [0.00, 0.00] | 0.66 [0.47, 0.82] |
| 11 | 0.54 [0.49, 0.59] | 0.04 [0.01, 0.07] | -0.01 [-0.06, 0.04] | -0.01 [-0.05, 0.03] | -0.00 [-0.02, 0.02] | 0.08 [0.01, 0.15] | 0.00 [0.00, 0.00] | 0.42 [0.32, 0.50] |
| 12 | 0.83 [0.78, 0.87] | 0.02 [-0.02, 0.06] | 0.00 [-0.05, 0.07] | -0.01 [-0.04, 0.02] | -0.01 [-0.05, 0.03] | 0.00 [-0.03, 0.04] | 0.00 [0.00, 0.01] | 0.70 [0.60, 0.80] |
| 13 | 0.98 [0.97, 0.99] | 0.01 [0.00, 0.02] | -0.02 [-0.06, 0.01] | -0.00 [-0.03, 0.02] | -0.00 [-0.04, 0.04] | -0.00 [-0.01, 0.01] | 0.00 [0.00, 0.00] | 0.95 [0.89, 0.99] |
| 14 | 0.36 [0.23, 0.49] | 0.00 [0.00, 0.00] | -0.02 [-0.04, 0.00] | -0.01 [-0.02, 0.00] | 0.00 [-0.00, 0.00] | -0.02 [-0.05, 0.01] | 0.00 [0.00, 0.00] | 0.36 [0.18, 0.55] |
| 15 | 0.95 [0.94, 0.97] | -0.02 [-0.06, 0.02] | 0.09 [0.02, 0.18] | 0.08 [-0.00, 0.18] | 0.08 [-0.01, 0.18] | -0.00 [-0.02, 0.02] | -0.00 [-0.04, 0.03] | 0.97 [0.95, 0.98] |
| 16 | 0.63 [0.57, 0.69] | 0.00 [-0.03, 0.03] | -0.01 [-0.04, 0.03] | -0.01 [-0.05, 0.03] | -0.02 [-0.05, 0.00] | -0.03 [-0.06, 0.01] | 0.00 [0.00, 0.00] | 0.55 [0.49, 0.61] |
| 17 | 0.46 [0.38, 0.55] | -0.02 [-0.05, -0.00] | 0.00 [-0.02, 0.03] | 0.00 [-0.01, 0.01] | -0.01 [-0.03, 0.00] | -0.05 [-0.09, -0.00] | 0.00 [0.00, 0.00] | 0.43 [0.30, 0.56] |
| 18 | 0.93 [0.90, 0.96] | 0.02 [-0.03, 0.08] | -0.03 [-0.06, -0.01] | -0.01 [-0.04, 0.02] | -0.03 [-0.07, 0.01] | 0.01 [-0.01, 0.04] | -0.04 [-0.09, 0.01] | 0.88 [0.81, 0.94] |
| 19 | 0.00 [0.00, 0.00] | 0.00 [0.00, 0.00] | 0.00 [0.00, 0.00] | 0.00 [0.00, 0.00] | 0.00 [0.00, 0.00] | 0.00 [0.00, 0.00] | 0.00 [0.00, 0.00] | 0.00 [0.00, 0.00] |
| mean | 0.73 [0.72, 0.75] | 0.02 [0.00, 0.03] | 0.02 [-0.00, 0.05] | 0.00 [-0.01, 0.02] | 0.01 [-0.01, 0.02] | -0.00 [-0.01, 0.01] | 0.05 [0.04, 0.06] | 0.72 [0.71, 0.74] |

Table 7: Forgetting for each method (columns) and tasks (rows).

| Method / Task | Fine-tuning | L2 | EWC | MAS | VCL | PackNet | Perfect Memory | A-GEM | mean |
|---|---|---|---|---|---|---|---|---|---|
| 0 | 0.13 [0.05, 0.20] | 0.14 [0.07, 0.21] | 0.13 [0.05, 0.20] | 0.13 [0.05, 0.20] | 0.02 [-0.09, 0.12] | -0.49 [-0.65, -0.34] | 0.43 [0.31, 0.54] | 0.13 [0.04, 0.22] | 0.08 |
| 1 | 0.06 [-0.09, 0.18] | -0.39 [-0.53, -0.25] | -0.10 [-0.19, -0.02] | -0.63 [-0.72, -0.54] | -0.51 [-0.60, -0.40] | -0.08 [-0.19, 0.03] | -0.68 [-0.83, -0.52] | 0.21 [0.14, 0.28] | -0.26 |
| 2 | 0.39 [0.23, 0.52] | -0.09 [-0.40, 0.19] | 0.35 [0.20, 0.49] | 0.04 [-0.20, 0.24] | -0.09 [-0.36, 0.14] | 0.33 [0.09, 0.51] | -0.82 [-1.18, -0.51] | 0.48 [0.37, 0.58] | 0.07 |
| 3 | 0.76 [0.73, 0.79] | -0.26 [-0.53, -0.01] | 0.66 [0.51, 0.76] | -0.09 [-0.31, 0.13] | -0.33 [-0.58, -0.12] | 0.80 [0.76, 0.83] | -0.71 [-0.91, -0.53] | 0.81 [0.78, 0.83] | 0.21 |
| 4 | 0.08 [-0.05, 0.20] | -0.45 [-0.52, -0.38] | -0.36 [-0.45, -0.26] | -0.35 [-0.45, -0.25] | -0.48 [-0.54, -0.42] | -0.21 [-0.33, -0.09] | -0.49 [-0.55, -0.43] | 0.18 [0.05, 0.29] | -0.26 |
| 5 | 0.55 [0.50, 0.59] | -1.84 [-2.39, -1.29] | 0.14 [0.04, 0.24] | -0.44 [-0.59, -0.29] | -0.38 [-0.58, -0.20] | 0.05 [-0.22, 0.29] | -1.96 [-2.41, -1.49] | 0.45 [0.30, 0.57] | -0.43 |
| 6 | -0.11 [-0.24, 0.00] | -1.69 [-1.94, -1.44] | -0.81 [-0.98, -0.62] | -1.38 [-1.60, -1.17] | -1.81 [-1.97, -1.65] | 0.12 [0.04, 0.20] | -2.92 [-3.05, -2.80] | -0.27 [-0.38, -0.17] | -1.11 |
| 7 | 0.13 [0.04, 0.22] | -0.20 [-0.26, -0.15] | -0.10 [-0.19, -0.01] | -0.18 [-0.24, -0.12] | -0.10 [-0.17, -0.03] | 0.44 [0.34, 0.53] | -0.24 [-0.29, -0.19] | 0.28 [0.14, 0.42] | 0.00 |
| 8 | 0.44 [0.38, 0.50] | -0.91 [-1.45, -0.38] | 0.35 [0.21, 0.47] | -0.18 [-0.56, 0.14] | 0.12 [-0.17, 0.35] | 0.45 [0.29, 0.58] | -1.86 [-2.20, -1.53] | 0.41 [0.27, 0.54] | -0.15 |
| 9 | 0.77 [0.70, 0.81] | -0.68 [-1.20, -0.25] | -0.11 [-0.64, 0.32] | -0.51 [-1.05, -0.05] | -0.66 [-1.21, -0.17] | 0.80 [0.72, 0.86] | -2.44 [-3.05, -1.95] | 0.79 [0.73, 0.84] | -0.26 |
| 10 | -0.30 [-0.60, -0.01] | -0.49 [-0.82, -0.15] | -0.66 [-0.96, -0.36] | -0.41 [-0.74, -0.10] | -0.07 [-0.35, 0.19] | -0.41 [-0.74, -0.10] | -1.73 [-1.85, -1.63] | -0.23 [-0.67, 0.16] | -0.54 |
| 11 | -0.21 [-0.30, -0.12] | -0.77 [-0.90, -0.65] | -0.57 [-0.70, -0.45] | -0.71 [-0.81, -0.61] | -0.91 [-1.01, -0.80] | 0.08 [-0.02, 0.17] | -1.07 [-1.15, -1.00] | -0.43 [-0.61, -0.25] | -0.57 |
| 12 | 0.04 [-0.14, 0.20] | -0.74 [-1.09, -0.43] | -0.36 [-0.68, -0.08] | -0.67 [-1.00, -0.38] | -0.38 [-0.66, -0.13] | 0.42 [0.32, 0.50] | -1.76 [-2.11, -1.48] | -0.19 [-0.47, 0.06] | -0.45 |
| 13 | 0.53 [0.44, 0.60] | -0.59 [-0.79, -0.41] | 0.39 [0.25, 0.51] | -0.33 [-0.55, -0.13] | -0.44 [-0.67, -0.22] | 0.82 [0.79, 0.85] | -0.78 [-0.96, -0.64] | 0.56 [0.44, 0.67] | 0.02 |
| 14 | -0.19 [-0.31, -0.05] | -0.49 [-0.55, -0.43] | -0.47 [-0.53, -0.41] | -0.45 [-0.53, -0.37] | -0.49 [-0.55, -0.43] | -0.24 [-0.35, -0.13] | -0.49 [-0.55, -0.43] | -0.25 [-0.41, -0.09] | -0.38 |
| 15 | 0.48 [0.41, 0.56] | -1.52 [-1.99, -1.08] | 0.03 [-0.13, 0.19] | -1.05 [-1.40, -0.73] | -0.20 [-0.41, -0.01] | 0.17 [-0.00, 0.34] | -1.94 [-2.44, -1.44] | 0.36 [0.24, 0.48] | -0.46 |
| 16 | -0.79 [-0.96, -0.63] | -2.44 [-2.64, -2.26] | -1.83 [-2.07, -1.59] | -2.07 [-2.28, -1.85] | -2.50 [-2.68, -2.33] | -0.34 [-0.48, -0.19] | -2.92 [-3.05, -2.80] | -1.05 [-1.23, -0.86] | -1.74 |
| 17 | 0.02 [-0.07, 0.11] | -0.23 [-0.28, -0.18] | -0.20 [-0.26, -0.13] | -0.22 [-0.28, -0.17] | -0.22 [-0.28, -0.17] | -0.03 [-0.12, 0.06] | -0.24 [-0.29, -0.19] | -0.03 [-0.13, 0.09] | -0.14 |
| 18 | 0.52 [0.45, 0.58] | -0.74 [-1.23, -0.30] | -0.06 [-0.46, 0.28] | -0.67 [-1.19, -0.17] | 0.26 [-0.04, 0.49] | 0.44 [0.28, 0.58] | -2.24 [-2.57, -1.86] | 0.52 [0.46, 0.58] | -0.25 |
| 19 | 0.73 [0.66, 0.78] | -0.59 [-1.08, -0.18] | -0.16 [-0.71, 0.29] | -0.26 [-0.77, 0.13] | -0.45 [-1.02, 0.03] | 0.46 [0.12, 0.70] | -2.49 [-3.11, -1.98] | 0.68 [0.58, 0.75] | -0.26 |
| mean | 0.20 [0.17, 0.23] | -0.75 [-0.87, -0.65] | -0.19 [-0.25, -0.14] | -0.52 [-0.58, -0.48] | -0.48 [-0.56, -0.42] | 0.18 [0.14, 0.21] | -1.37 [-1.46, -1.30] | 0.17 [0.13, 0.20] | — |

Table 8: Forward transfers on CW20 (rows) for each method (columns).

| Task Method | $\Delta_0$ | $\Delta_1$ | $\Delta_2$ | $\Delta_3$ | $\Delta_4$ | $\Delta_5$ | $\Delta_6$ | $\Delta_7$ | $\Delta_8$ | $\Delta_9$ |
|---|---|---|---|---|---|---|---|---|---|---|
| Fine-tuning | -0.42 | -0.27 | -0.34 | -0.24 | -0.26 | -0.06 | -0.68 | -0.11 | 0.08 | -0.04 |
| L2 | -0.63 | -0.39 | -0.66 | -0.33 | -0.03 | 0.32 | -0.76 | -0.02 | 0.17 | 0.10 |
| EWC | -0.79 | -0.47 | -0.71 | -0.27 | -0.11 | -0.11 | -1.02 | -0.09 | -0.41 | -0.05 |
| MAS | -0.54 | -0.08 | -0.71 | -0.24 | -0.10 | -0.61 | -0.68 | -0.04 | -0.49 | 0.25 |
| VCL | -0.08 | -0.40 | -0.29 | -0.11 | -0.01 | 0.18 | -0.69 | -0.13 | 0.13 | 0.20 |
| PackNet | 0.09 | 0.16 | 0.08 | 0.02 | -0.03 | 0.12 | -0.46 | -0.47 | -0.01 | -0.34 |
| Perfect Memory | -2.16 | -0.39 | -0.94 | -0.07 | 0.00 | 0.02 | 0.00 | 0.00 | -0.38 | -0.05 |
| A-GEM | -0.36 | -0.63 | -0.67 | -0.25 | -0.43 | -0.09 | -0.77 | -0.32 | 0.11 | -0.11 |

Table 9: Difference in transfer when revisiting tasks. $\Delta_i = \text{FT}_{i+10} - \text{FT}_i$; recall that $i$ and $i + 10$ are the same tasks in CW20.

| method | performance | forgetting | f. transfer |
|---|---|---|---|
| **Fine-tuning** | 0.10 [0.10, 0.11] | 0.74 [0.72, 0.75] | **0.32** [0.28, 0.35] |
| **L2** | 0.48 [0.43, 0.53] | 0.02 [-0.00, 0.04] | -0.57 [-0.77, -0.39] |
| **EWC** | 0.66 [0.62, 0.69] | 0.03 [0.01, 0.06] | 0.05 [-0.02, 0.12] |
| **MAS** | 0.59 [0.56, 0.61] | -0.02 [-0.03, -0.00] | -0.35 [-0.42, -0.28] |
| **VCL** | 0.53 [0.49, 0.58] | -0.02 [-0.03, -0.00] | -0.44 [-0.57, -0.32] |
| **PackNet** | **0.83** [0.81, 0.85] | -0.00 [-0.01, 0.01] | 0.21 [0.16, 0.25] |
| **Perfect Memory** | 0.29 [0.26, 0.31] | 0.03 [0.02, 0.05] | -1.11 [-1.20, -1.04] |
| **A-GEM** | 0.13 [0.12, 0.14] | 0.73 [0.72, 0.75] | **0.29** [0.27, 0.32] |
| **MT** | 0.51 [0.48, 0.53] | — | — |
| **MT (PopArt)** | 0.65 [0.63, 0.67] | — | — |

Table 10: Results on CW10, for CL methods and multi-task training.

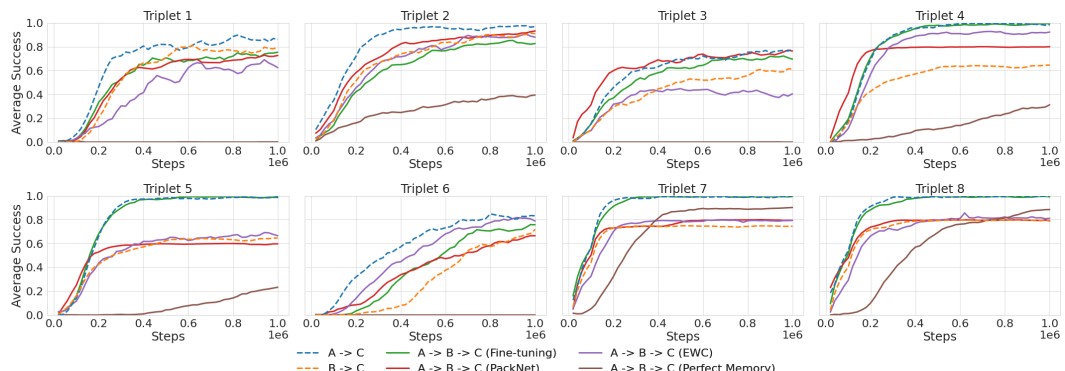

Figure 11: Performance of selected methods on the triplets sequences.

| Triplet | A→ C | B → C | A → B → C (Fine-tuning) | A → B → C (Pack-Net) | A → B → C (EWC) | A → B → C (Perfect Mem.) |
|---------|------|-------|-------------------------|----------------------|-----------------|---------------------------|
| 1 | 0.20 | -0.10 | -0.14 | -0.21 | -0.50 | -1.73 |
| 2 | 0.64 | 0.24 | 0.06 | 0.36 | 0.20 | -0.98 |
| 3 | 0.21 | -0.15 | 0.10 | 0.28 | -0.31 | -1.09 |
| 4 | 0.76 | 0.20 | 0.76 | 0.56 | 0.62 | -0.52 |
| 5 | 0.75 | 0.20 | 0.76 | 0.20 | 0.22 | -0.64 |
| 6 | 0.42 | -0.02 | 0.14 | 0.10 | 0.29 | -0.49 |
| 7 | 0.80 | -0.01 | 0.79 | 0.11 | 0.06 | -0.00 |
| 8 | 0.79 | 0.10 | 0.74 | 0.15 | 0.05 | -0.46 |
| mean | 0.57 | 0.06 | 0.40 | 0.19 | 0.08 | -0.74 |

Table 11: Forward Transfers for triplets.

# F  Triplet experiments

In addition to the CW10 and CW20 sequences, we propose eight triplets. They are sequences of three tasks aimed to allow rapid experimenting. The triplets were selected in order to capture situations when remembering is crucial for forward transfer. That is, we consider tasks $A \rightarrow B \rightarrow C$, where $A \rightarrow C$ exhibits bigger forward transfer than $B \rightarrow C$ (based on the transfer matrix), see Table 11. We propose the following triplets:

1. `push-v1` $\rightarrow$ `window-close-v1` $\rightarrow$ `hammer-v1`

2. `hammer-v1` $\rightarrow$ `window-close-v1` $\rightarrow$ `faucet-close-v1`

3. `stick-pull-v1` $\rightarrow$ `push-back-v1` $\rightarrow$ `push-wall-v1`

4. `push-wall-v1` $\rightarrow$ `shelf-place-v1` $\rightarrow$ `push-back-v1`

5. `faucet-close-v1` $\rightarrow$ `shelf-place-v1` $\rightarrow$ `push-back-v1`

6. `stick-pull-v1` $\rightarrow$ `peg-unplug-side-v1` $\rightarrow$ `stick-pull-v1`

7. `window-close-v1` $\rightarrow$ `handle-press-side-v1` $\rightarrow$ `peg-unplug-side-v1`

8. `faucet-close-v1` $\rightarrow$ `shelf-place-v1` $\rightarrow$ `peg-unplug-side-v1`

In Figure 11 and Table 11 we present evaluations for the third task. On most of the triplets, the tested methods (Fine-tuning, EWC, PackNet, Perfect Memory) are not able to reach the performance obtained by direct transfer $A \rightarrow C$. Moreover, Fine-tuning often outperforms the rest of the methods. Interestingly, in triplets 7 and 8, Fine-tuning is able to achieve transfer comparable to $A \rightarrow C$, while EWC and PackNet smaller one similar to $B \rightarrow C$. Perfect memory does not perform well on any triplet, which is consistent with our experiments on the long sequences.

## G Ablations

### G.1 Other orders of tasks

We study the impact of task order on the performance of CL methods. In Figure 12, we show average performance curves for three selected CL methods that are run with the same hyperparameters on three different orderings: the CW10 sequence (see Section A.2) and its two random permutations. Further, Table 12 contains values of the forgetting and forward transfer metrics.

The ranking of the methods is preserved; we also observe that forward transfer of EWC varies substantially with the change of sequence order. To a lesser extent, this also impacts the performance metric. Decreasing this variance is, in our view, an important research question. We speculate that it is much related to improving transfer in general.

| method | performance | forgetting | f. transfer |
|---|---|---|---|
| **Fine-tuning, CW10 order (standard)** | 0.10 [0.10, 0.11] | 0.74 [0.72, 0.75] | 0.32 [0.29, 0.35] |
| **EWC, CW10 order (standard)** | 0.66 [0.63, 0.69] | 0.03 [0.01, 0.06] | 0.02 [-0.05, 0.07] |
| **PackNet, CW10 order (standard)** | 0.87 [0.85, 0.89] | 0.01 [-0.01, 0.02] | 0.38 [0.36, 0.41] |
| **Fine-tuning, CW10 permutation 1** | 0.10 [0.10, 0.10] | 0.72 [0.69, 0.74] | 0.18 [0.13, 0.23] |
| **EWC, CW10 permutation 1** | 0.46 [0.43, 0.50] | 0.02 [0.00, 0.04] | -0.45 [-0.50, -0.41] |
| **PackNet, CW10 permutation 1** | 0.82 [0.80, 0.84] | 0.03 [0.01, 0.05] | 0.25 [0.19, 0.30] |
| **Fine-tuning, CW10 permutation 2** | 0.10 [0.10, 0.10] | 0.73 [0.72, 0.75] | 0.22 [0.19, 0.26] |
| **EWC, CW10 permutation 2** | 0.48 [0.43, 0.52] | 0.05 [0.03, 0.07] | -0.54 [-0.68, -0.42] |
| **PackNet, CW10 permutation 2** | 0.84 [0.81, 0.86] | 0.01 [0.00, 0.03] | 0.30 [0.26, 0.34] |

Table 12: Results on the CW10 sequence and its 2 random permutations.

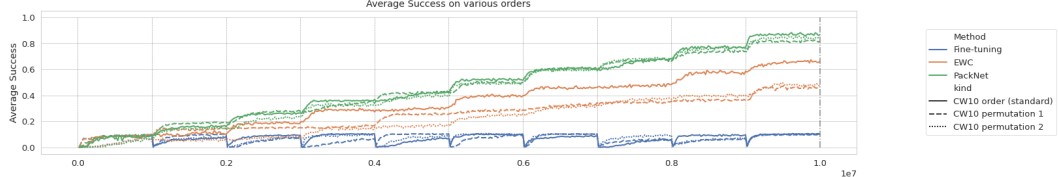

Figure 12: Average performance for selected methods on CW10 and its two random orderings.

The new sequences are:

```
handle-press-side-v1, faucet-close-v1, shelf-place-v1, stick-pull-v1,
peg-unplug-side-v1, hammer-v1, push-back-v1, push-wall-v1, push-v1, window-close-v1
```

and

```
stick-pull-v1, push-wall-v1, shelf-place-v1, window-close-v1, hammer-v1,
peg-unplug-side-v1, push-back-v1, faucet-close-v1, push-v1, handle-press-side-v1.
```

### G.2 Random order of tasks

We further study the impact of task orderings by evaluating on 20 different random orders, see Table 13. Obtained results are qualitatively similar to the ones with the fixed ordering.

### G.3 One-hot inputs

We study an alternative setup in which a single-head architecture is used, and one-hot encoding is provided in input for the task identification. In Table 14 and Figure 13 we present results for three selected CL methods in both the multi-head and single-head setups. We see that the performance of EWC drops a bit, while PackNet remains mostly unaffected. Table 14 contains values of average performance, forgetting, and forward transfer computed in this setting.

| method | performance | forgetting | f. transfer |
|---|---|---|---|
| **Fine-tuning** | 0.08 [0.07, 0.09] | 0.71 [0.68, 0.74] | 0.17 [0.09, 0.24] |
| **L2** | 0.42 [0.34, 0.49] | 0.00 [-0.01, 0.01] | -0.73 [-0.90, -0.56] |
| **EWC** | 0.53 [0.48, 0.58] | 0.01 [-0.01, 0.03] | -0.31 [-0.43, -0.18] |
| **MAS** | 0.41 [0.35, 0.47] | 0.00 [-0.02, 0.02] | -0.65 [-0.79, -0.51] |
| **VCL** | 0.39 [0.34, 0.44] | 0.02 [-0.00, 0.03] | -0.69 [-0.80, -0.58] |
| **PackNet** | 0.72 [0.68, 0.76] | 0.00 [-0.01, 0.01] | -0.04 [-0.14, 0.05] |
| **Perfect Memory** | 0.27 [0.24, 0.31] | -0.00 [-0.02, 0.01] | -1.12 [-1.22, -1.02] |
| **A-GEM** | 0.10 [0.08, 0.11] | 0.70 [0.67, 0.73] | 0.19 [0.12, 0.24] |

Table 13: Results averaged over 20 different random orderings of CW10 – see Subsection G.2

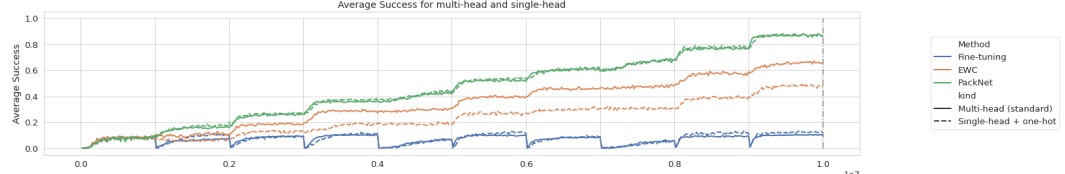

Figure 13: Average performance for selected CL methods on CW10 with standard multi-head architecture and single-head architecture with one-hot task encoding added to input.

For the single-head experiments, we ran another hyperparameter search. For the single-head EWC experiment the selected value is $\lambda = 10^3$ as opposed to $\lambda = 10^4$ for multi-head. We conjecture that the multi-head case can be regularized more aggressively, as for each task, it gets a small number of new parameters (in new heads).

| method | performance | forgetting | f. transfer |
|---|---|---|---|
| **Fine-tuning, Multi-head (standard)** | 0.10 [0.10, 0.11] | 0.74 [0.72, 0.75] | 0.32 [0.29, 0.35] |
| **EWC, Multi-head (standard)** | 0.66 [0.63, 0.69] | 0.03 [0.01, 0.06] | 0.02 [-0.05, 0.07] |
| **PackNet, Multi-head (standard)** | 0.87 [0.85, 0.89] | 0.01 [-0.01, 0.02] | 0.38 [0.36, 0.41] |
| **Fine-tuning, Single-head + one-hot** | 0.12 [0.12, 0.13] | 0.68 [0.66, 0.70] | 0.16 [0.12, 0.19] |
| **EWC, Single-head + one-hot** | 0.48 [0.45, 0.50] | 0.14 [0.12, 0.15] | -0.25 [-0.33, -0.17] |
| **PackNet, Single-head + one-hot** | 0.87 [0.85, 0.89] | -0.01 [-0.02, 0.01] | 0.24 [0.20, 0.28] |

Table 14: Results on CW10 with original multi-head setup and alternative with single-head and one-hot encoding

### G.4 Experiments with a sequence of 30 tasks

An important question for continual learning is how methods scale to a larger number of tasks. To test this in our setting, we consider a sequence of 30 tasks. To select challenging but solvable tasks we removed 10 easiest and 10 hardest task of MetaWorld (measured by performance in single-task learning).

This way we obtained a set of 30 following tasks: PLATE-SLIDE-V1, PLATE-SLIDE-BACK-SIDE-V1, HANDLE-PRESS-V1, HANDLE-PULL-V1, HANDLE-PULL-SIDE-V1, SOCCER-V1, COFFEE-PUSH-V1, COFFEE-BUTTON-V1, SWEEP-INTO-V1, DIAL-TURN-V1, HAND-INSERT-V1, WINDOW-OPEN-V1, PLATE-SLIDE-SIDE-V1, PLATE-SLIDE-BACK-V1, DOOR-LOCK-V1, DOOR-UNLOCK-V1, PUSH-V1, DOOR-OPEN-V1, BOX-CLOSE-V1, FAUCET-OPEN-V1, COFFEE-PULL-V1, SHELF-PLACE-V1, FAUCET-CLOSE-V1, HANDLE-PRESS-SIDE-V1, PUSH-WALL-V1, SWEEP-V1, STICK-PUSH-V1, BIN-PICKING-V1, BASKETBALL-V1, HAMMER-V1.

We then train four methods: fine-tuning, EWC, PackNet, and Perfect Memory on this set of tasks. We use the random ordering approach presented in Subsection G.2 with 40 seeds to obtain reliable results. The tested methods perform similarly as in the case of CW20, see Table 15. PackNet significantly outperforms the rest of the methods, followed by EWC. Reservoir and fine-tuning are affected with catastrophic forgetting, with performance showing that they are mostly able to solve a single task.

|  | Fine-tuning | EWC | PackNet | Perfect Memory |
|---|---|---|---|---|
| **Performance** | 0.04 [0.03, 0.04] | 0.44 [0.42, 0.46] | 0.70 [0.68, 0.71] | 0.04 [0.03, 0.04] |

Table 15: Results on 30 tasks.

## H   Multi-task learning

In multi-task experiments, we train 10 tasks from CW10 simultaneously (we do not duplicate tasks as it is done in CW20). In our experiments, PopArt reward normalization [21] yields significant improvements (0.66 vs 0.50 for the standard version, see also the top graph in Figure 14). We observe that EWC and PackNet achieve results similar or better to multi-task learning. We consider this to be an interesting research direction. We conjecture that CL methods might benefit from the fact that the critic network stores only one value function (corresponding to the current task), see also discussion in Section C.1.

For multi-task training, we tested the following hyperparameter values: batch size $\in \{128, 256, 512\}$ (selected value = 128), learning rate $\in \{3 \times 10^{-5}, 1 \times 10^{-4}, 3 \times 10^{-4}, 1 \times 10^{-3}\}$ (selected value $= 1 \times 10^{-4}$).

## I   Infrastructure used

We ran our experiments on clusters with servers typically equipped with 24 or 28 CPU cores and 64GB of memory (no GPU). A typical experiment CW20 experiment was 100 hours long and used 8 or 12 CPU cores.

During the project, we run more than 45K experiments.

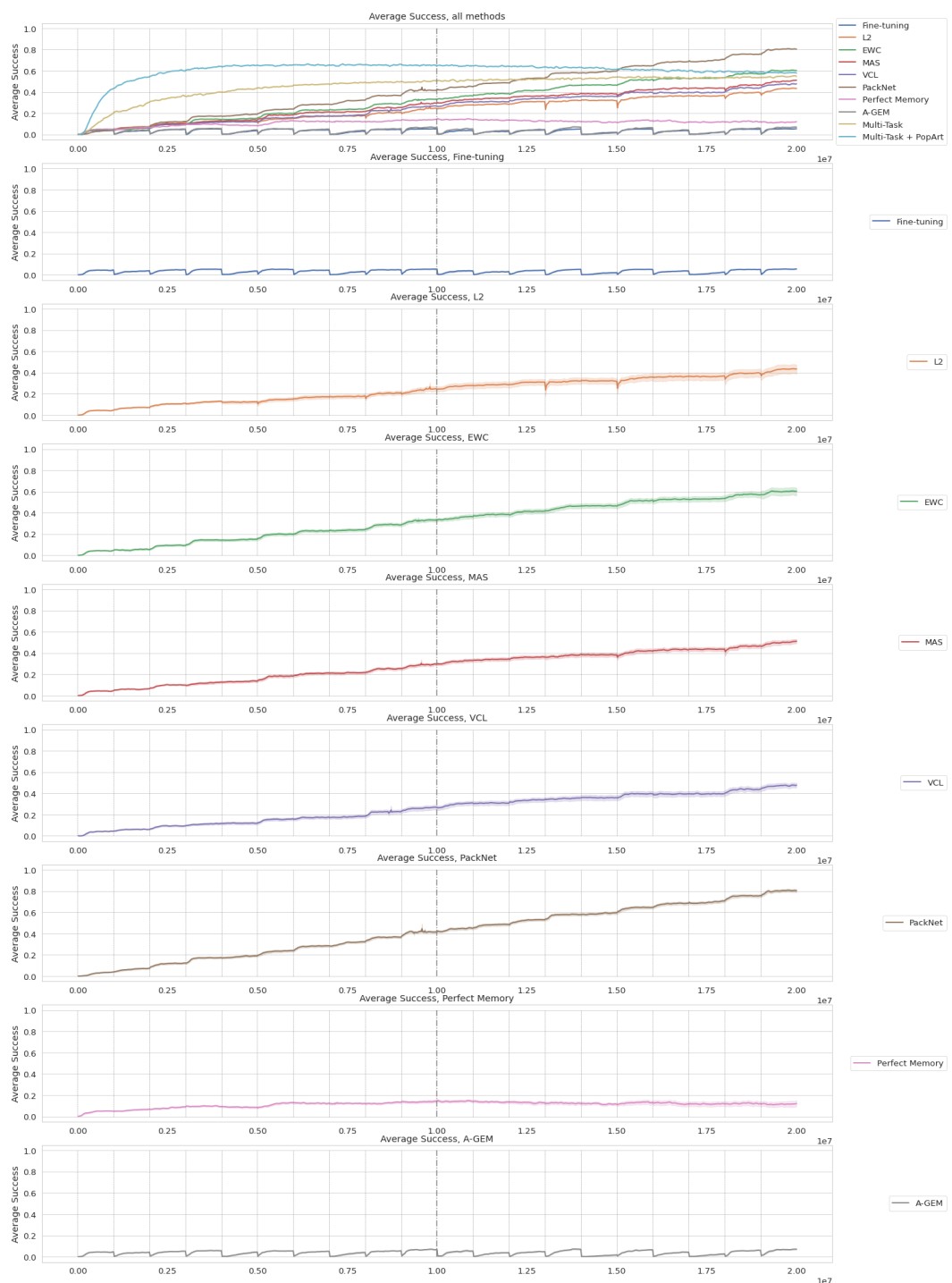

Figure 14: Average (over tasks) success rate for all tested methods and multi-task training.

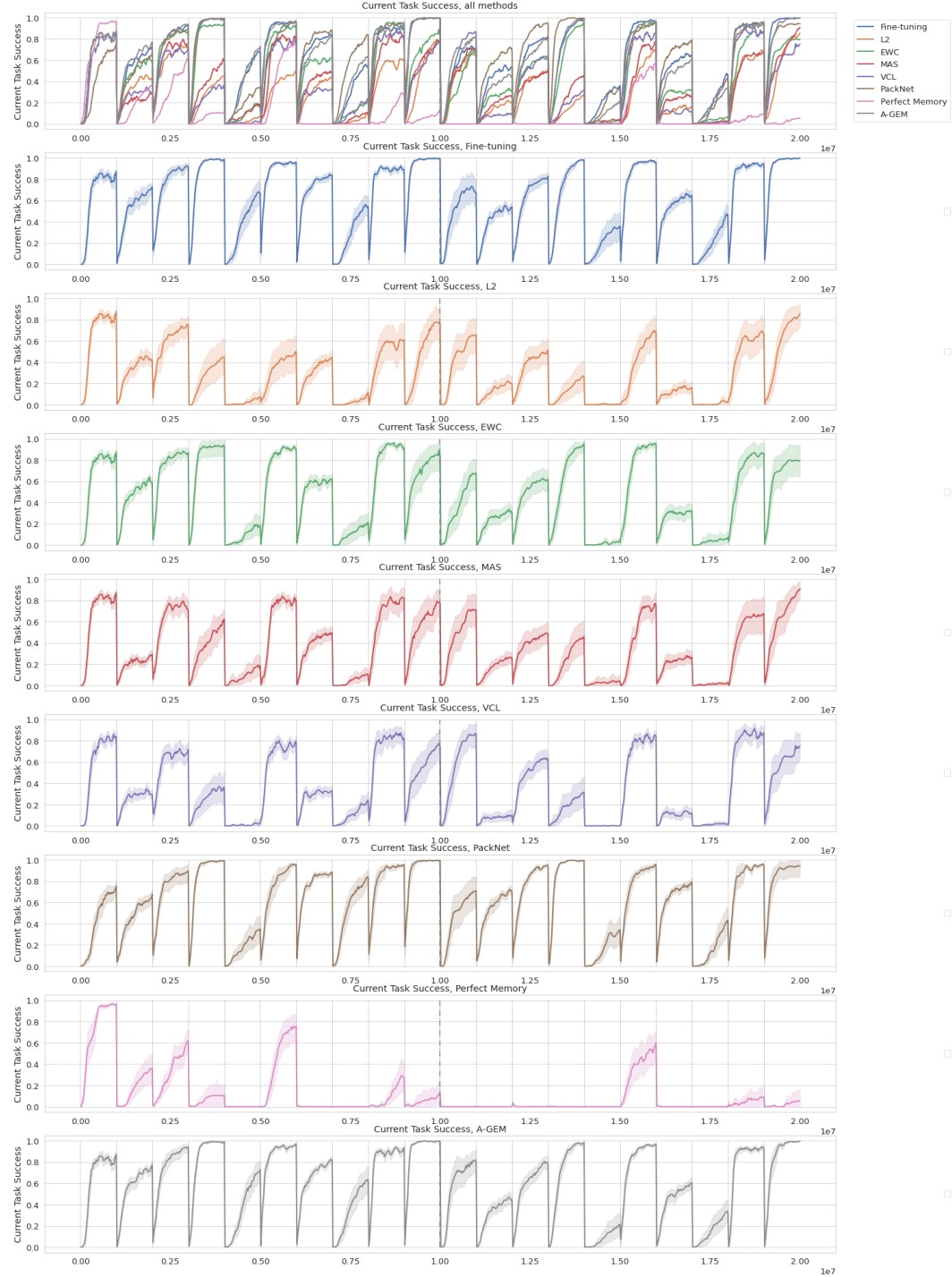

Figure 15: Success rate for the current task (the one being trained).

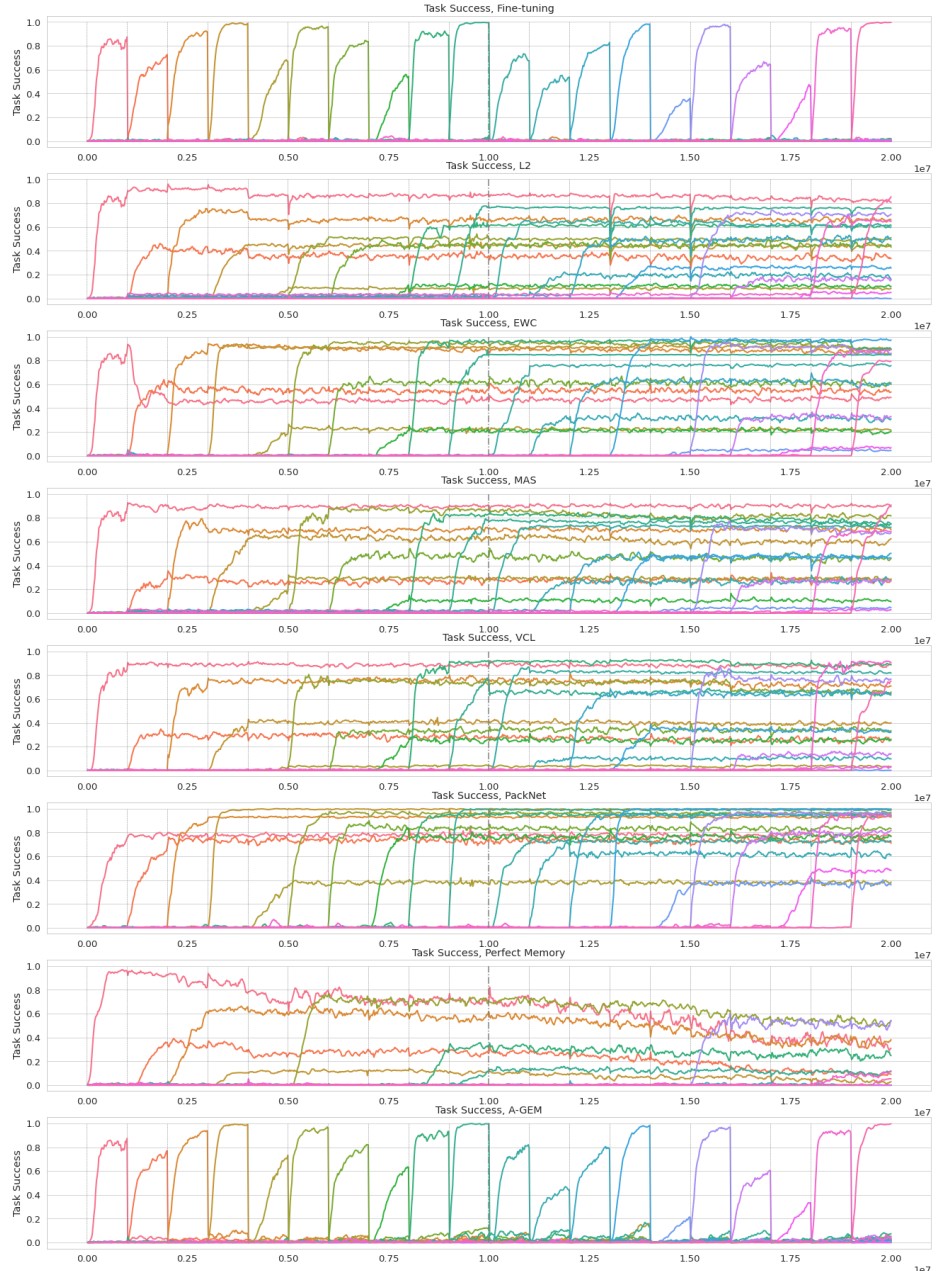

Figure 16: Success rates for all tasks.

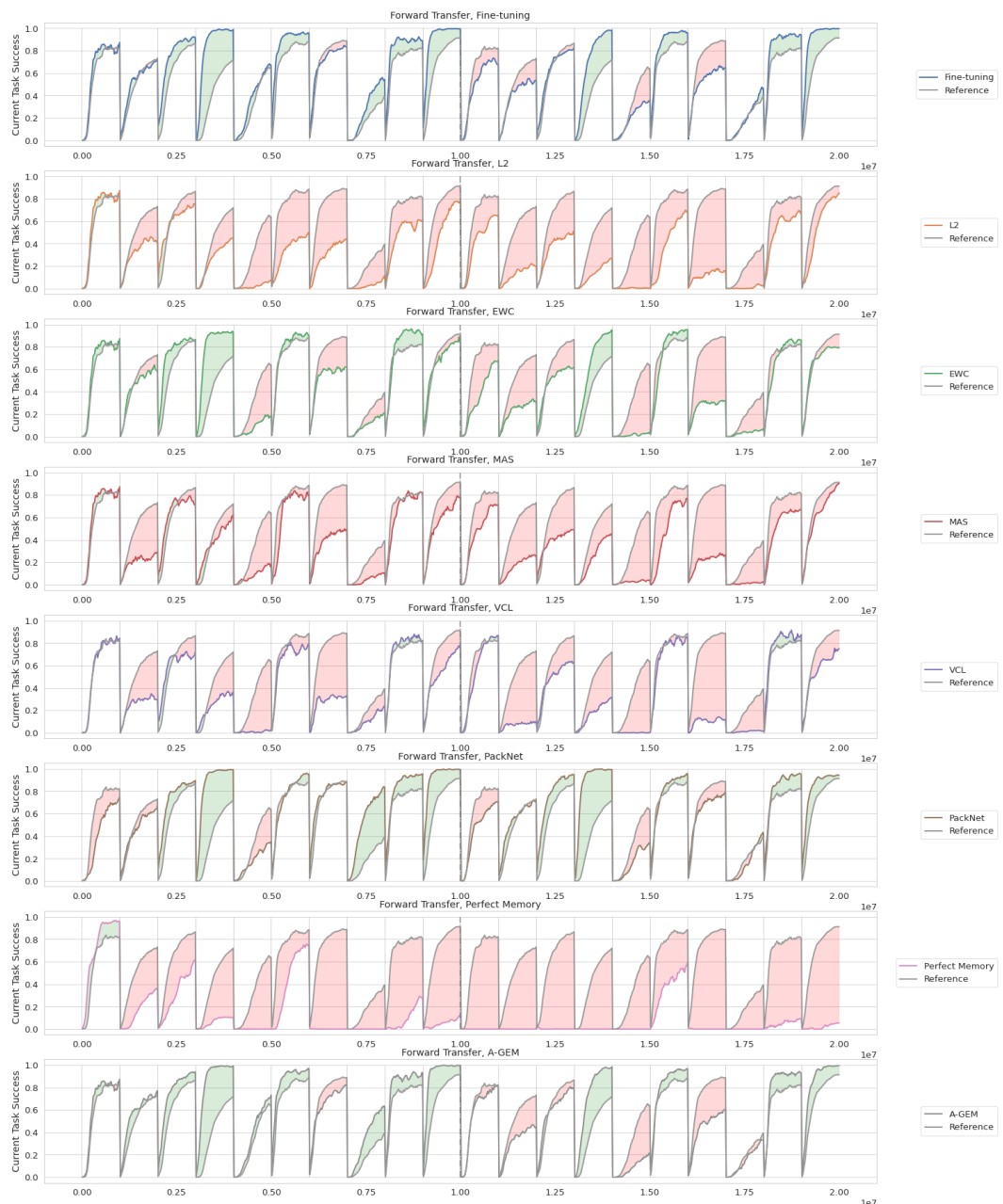

Figure 17: Forward transfer, the reference curves come from the single task training.

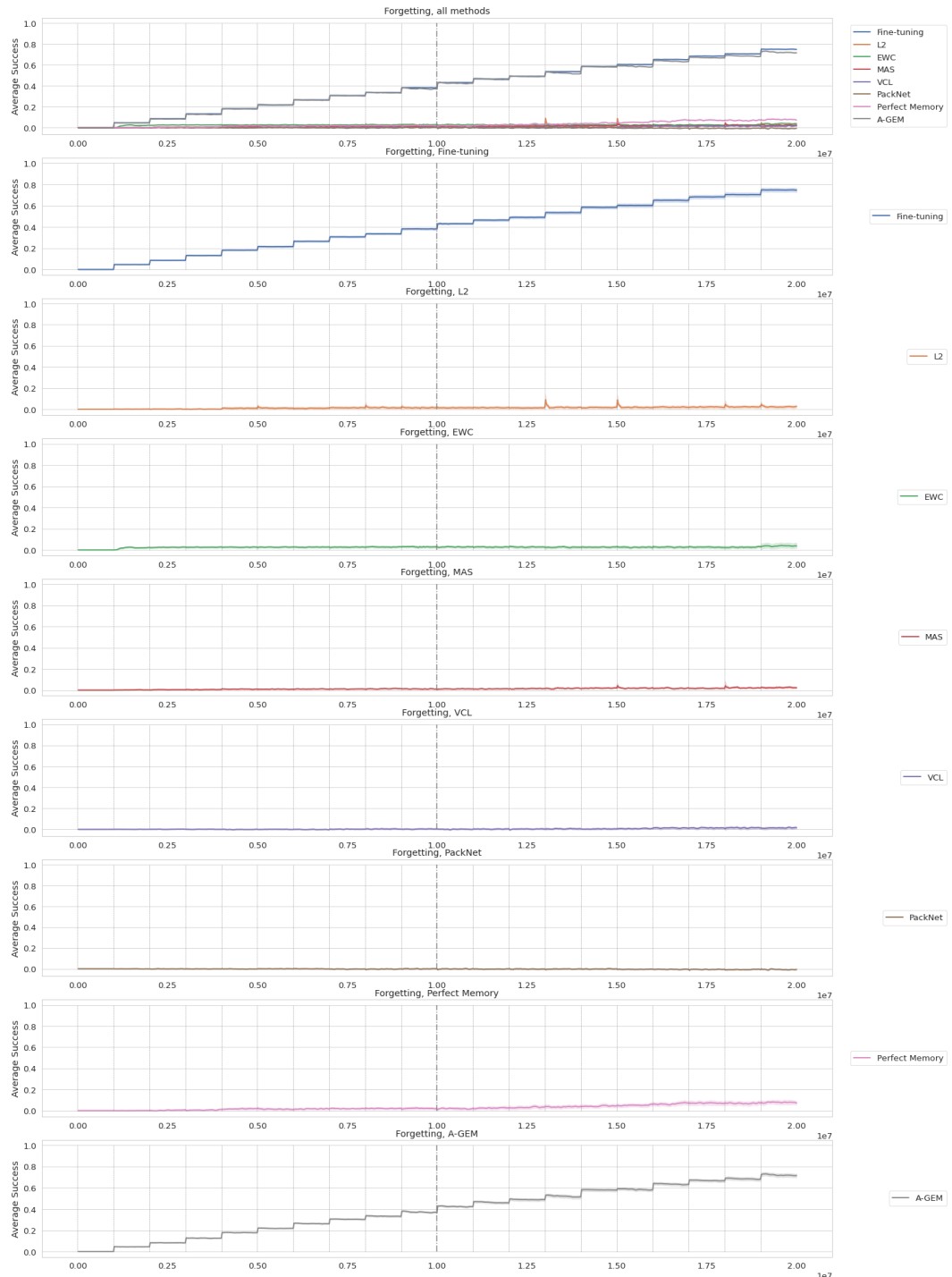

Figure 18: Average (over task) forgetting. For the task $i$ we set forgetting at time $t$ to be $\mathrm{F}_i(t) := p_i(i \cdot \Delta) - p_i(t)$ if $t \geq i \cdot \Delta$ and 0 otherwise. See also (3).