# OpenReview forum: "Continual World: A Robotic Benchmark For Continual Reinforcement Learning"
_NeurIPS.cc/2021/Conference — NeurIPS 2021 Poster_

### Official Review · Reviewer_EkWJ · 2021-07-13

**Rating:** 6
**Confidence:** 3

**Summary:**

The paper presents a suite of continuous control tasks for studying continual reinforcement learning. The paper includes a comprehensive study evaluating prior continual learning methods, both the ones designed for RL and for other tasks. It also highlights the challenges of applying existing RL methods to continual learning setting.

**Limitations And Societal Impact:**

The "limitation" section is adequate for technical contributions. It'd be interesting to see a short discussion on societal impact of studying the continual RL setting.

**Main Review:**

Key contribution
- A suite of continuous control tasks for continual reinforcement learning (CRL) (built on Meta World tasks)
- Implementations of existing continual learning (CL) methods, both the ones designed for RL and other tasks --- require non trivial adaptation.
- A fairly comprehensive study using existing methods on the proposed benchmark.
- Highlights CRL-related challenges (Sec. 6.1) --- 1) existing actor-critic methods may not be the best for CRL settings and 2) we may need to rethink the standard RL training procedure, i.e., exploration and how to populate replay buffers, in the CL setting.

Comments
- From my understanding, the key selling point of the proposed benchmark is “more realistic robotics tasks”, but practically, I’m not sure if they are “realistic” in the sense that they are still individual tasks with clear assumptions about task boundaries. Learning is harder for sure, but I’m not sure about realism.
- In addition, I'm not sure about why the previous supervised CL benchmarks “are not geared towards measuring forward transfer” (line 65). As far as I could tell, the forward transfer measures the learning speed / sample efficiency as a result of transferring the knowledge from the previous task. It would seem natural to also measure the sample efficiency as a form of forward transfer in supervised CL?
- It would be interesting to see what the harder tasks bring to the table. Ideally, the paper should conduct a small-scale experiment and apply the same methods on “easier” tasks such as MNIST sequence and see if the conclusion still holds, since the main selling point is that the easier tasks cannot show the challenge of “more realistic settings”
- Triplet experiment: ideally PackNet should not be affected by the prior tasks (i.e., task B) since it isolates parameters for each task, but it doesn’t seem to be the case in this experiment. More analysis would be helpful.


Minor:
- Fig 1 no axis
- Including Multi-task learning + PopArt in Fig 3 as a curve is somewhat confusing, since the training curve means an entirely different thing here (learning all tasks simultaneously). Better include the asymptotic performance as a horizontal line?


**Time Spent Reviewing:**

4

---

> ### Author Response · Authors · 2021-08-10
> **Answer to Rev EkWJ**
>
> Thank you for your review.
>
> We should have been more precise with wording. We meant that our benchmark has tasks evaluation protocols that reflect realistic use cases. It is why we have based on MetaWorld, which was designed with this goal in mind. We believe that our benchmark is a valid step in this direction, although there are many limitations that we list in our work and extensions that we hope to do as future work. We discuss the explicit task boundaries in the limitation section (sec 4.4), which we will further extend. We also note that our primary focus is to direct the community research attention to the interaction between forgetting/forward transfer and how improving forgetting can help transfer (orthogonal to the task boundary question). Doing task boundary detection (or dealing with an alternative formulation of the problem) would also be problematic when working from states. It would make the problem trivial (e.g., if task id is part of the observation) or impossible. Moving to visual observation would allow studying the task boundary detection problem. However, as we mentioned already (in answer to Rev Mh4F and the limitation section), it would considerably increase the computational needs (and wall clock time) needed to run experiments, making the benchmark less accessible.
>
> We agree (and argue) that forward transfer is understudied in current CL research (with some notable exceptions [put some papers]). We note that supervised CL and RL CL exhibit different characteristics. For example, replay methods excel in supervised learning (see e.g.  [1, 2, 3, 4]) but underperform in our robotic tasks.  For the sake of clarity, in this (already packed) paper, we decided to focus solely on reinforcement learning tasks. We believe there are significantly more benchmarks for the supervised scenario than the RL ones hence our focus on RL. Due to lack of space, we could not fit a careful analysis of this setting along with the RL setting. We note, however, that a key component to the proposed setting is that in a sequence, some tasks act as ‘’distractors’’. A good supervised learning scenario for this work should ensure that certain tasks do end up leading to a significant reduction of forward transfer. This is not a given for tasks that are so highly related (like split CIFAR), so it might be that even in the supervised setting, a more challenging benchmark is required to highlight the relationship between transfer and forgetting properly.
>
> Concerning PackNet (in triplet experiments), we note that even though the parameters are isolated/frozen, there is interaction via the activations (activations related to the previous tasks are not masked and can be reused). We will provide a more detailed description in the camera-ready version.
>
> We also thank you for the minor comments, which we will take into account.
>
> [1] Chaudhry et al. “On Tiny Episodic Memories in Continual Learning.” ArXiv:1902.10486, 2019.
>
> [2] Ven, Gido M. van de, and Andreas S. Tolias. “Three Scenarios for Continual Learning.” ArXiv Preprint ArXiv:1904.07734, 2019.
>
> [3] Hsu, Yen-Chang, et al. “Re-Evaluating Continual Learning Scenarios: A Categorization and Case for Strong Baselines.” ArXiv:1810.12488, 2018.
>
> [4] Buzzega et al. “Dark Experience for General Continual Learning: A Strong, Simple Baseline.” 34th NeurIPS 2020.

---

### Official Review · Reviewer_Mh4F · 2021-07-15

**Rating:** 6
**Confidence:** 3

**Summary:**

Summary. The paper proposes a robotic benchmark for continual reinforcement learning built on top of meta-world benchmark. It analyzes the difficulty in the problem of continual RL (such as catastrophic forgetting) and proposes metrics (average performance, forward transfer, forgetting) to measure performance of continual RL algorithms. Additionally, the paper evaluates existing continual RL algorithms on their benchmark and discusses the limitations of the proposed benchmark in detail as well.

**Limitations And Societal Impact:**

It's not discussed in the paper

**Main Review:**

Originality: The paper doesn’t propose any new method but rather presents a new robotic continual RL benchmark.

Quality: The paper appropriately analyzes the proposed benchmark, evaluates existing methods on it and discusses the limitations of their proposed benchmark in detail. I do think that absence of visual continual RL tasks makes the benchmark limiting as it inhibits study of interesting forms of transfer involving visual features.

Clarity: The paper is clearly written.

Significance: The paper is important because it provides a robotic benchmark for continual RL along with evaluating existing methods on the proposed benchmark which makes it easier to test new continual RL algorithms.

**Time Spent Reviewing:**

1 hr

---

> ### Author Response · Authors · 2021-08-10
> **Answer to Rev Mh4F**
>
> Thank you for your review.
>
> Using the state input was a conscious (but not easy) choice with the aim to balance being computationally accessible and meaningful. The CL setting requires learning tasks sequentially. When using states, the proposed tasks can be learned relatively fast, such that putting 20 of them is not prohibitive. Further, our work is meant to showcase RL-specific issues for CL. We opt for the most common RL setting in continuous control (i.e., state-based). This makes it possible to build on extensive existing knowledge, starting from very practical issues (e.g., what algorithms work, how to set them and tune) to inspiration from related fields (e.g., adapting meta-RL algorithms developed in the context of MetaWorld).
>
> Having said that, we acknowledge that having visual input is an important research direction which we make explicit in the limitation section; Sec 4.4). We plan to deliver the pixel-based version in the future.

---

### Official Review · Reviewer_7Se4 · 2021-07-16

**Rating:** 6
**Confidence:** 4

**Summary:**

This paper proposes a new benchmark for continual reinforcement learning. It modifies the Meta-World robotics benchmark that’s designed for meta- and multi-task RL evaluation by learning the tasks in sequence rather than together. This work is very timely as there lacks a common benchmark for continual/lifelong RL.

The benchmark consists of 10 tasks from Meta-World. To evaluate catastrophic forgetting, these 10 tasks are repeated for a sequence of length 20. For quicker iteration, there are also different triplets of tasks where the middle task might inhibit transfer to the last task.

**Limitations And Societal Impact:**

Yes. See main review for other limitations.

**Main Review:**

Originality:
- Continual World modifies the evaluation protocol of the existing Meta-World benchmark.

- As noted by the authors, there is already work that re-purposes Meta-World for the continual RL setting and evaluates various algorithms on it [33]. Additionally, [33] performs the evaluation on the full set of 50 tasks, while Continual World only looks at 10 tasks.

Quality:
- The 10 tasks in Continual World were carefully selected so that there is a good range in the difficulty of transfer between these tasks. They also include a transfer matrix, which can aid researchers when selecting the ordering of the tasks and understanding which tasks facilitate easier or harder transfer.

- They also evaluate a diverse range of continual learning methods on the benchmark. Interestingly, continuous fine-tuning gives the best forward transfer performance, but even that is far below the reference forward transfer value which is the highest transfer value from a single source task.

- Other methods I’d be curious to see the forward transfer performance of are: progressive neural networks and meta-RL methods adapted to the online setting.

Clarity:
- The paper is overall well-written and organized. The related work section also provides a thorough survey of the existing benchmarks.

Significance:
- A benchmark for continual RL is highly relevant to the NeurIPS community as there is no commonly used benchmark yet.

- The forward transfer results, when compared to the forgetting scores, really highlight transfer as an important direction to study in the context of continual RL.

Questions:
- In Table 1, is the Performance column reporting the success rate?

**Time Spent Reviewing:**

3

---

> ### Author Response · Authors · 2021-08-10
> **Answer to Rev 7Se4**
>
> Thank you for your encouraging review.
>
> Yes, the performance column (in Table 1) reports the success rate. More precisely, the average over tasks, calculated according to eq. (1). This will be clarified in the camera-ready version.
>
> [33] indeed uses MetaWorld tasks to evaluate CL methods (which we indicate in the related work section). However, their focus and scope are fundamentally different. Our goal was to provide a benchmark that allows us to look at the interaction between transfer and forgetting, and we set up a practical protocol with this in mind. Specific differences are as follows:
> * Contrary to [33], we use CW20, a subset of MetaWorld tasks. We evaluated all 50 tasks of MetaWorld during our research and found a considerable variance in difficulty, which would obfuscate the analysis. Instead, we carefully chose CW20, considering the mutual interaction (see the transfer matrix).
> * From the practical standpoint, using all 50 tasks would make experiments much longer (CL methods need to be trained sequentially). Contrary, we found that CW10, a shorter sequence of 10 tasks, is already meaningful and informative.
> * We use average success rate (instead of average return in [33]). This (technically easy) change is fundamental for our goals, as it enables us to calculate forgetting and forward transfer meaningfully. In fact, [33] does not introduce these metrics at all.
> * We consider a wider family of methods (e.g. parameter isolation-based, like PackNet) and carefully study various hyperparameters settings and the overall setup (e.g. regularizing the critic).
>
> As for the progressive networks, we had an internal discussion and decided against including them. The fundamental reason is that they are not truly scalable with the number of tasks (which conflicts with our desiderata for memory-efficient algorithms). Further, this model is similar to PackNet (in fact, PackNet is a more scalable version of progressive nets). We agree that evaluating meta-RL methods might be very interesting.
>
> We hope that these methods (and others) will be gradually included. Our benchmark is meant to be community-driven and open. We strongly encourage other researchers to contribute with the result of their/other methods. To facilitate this, we have prepared a website (not included here due to the double-blind review process).

---

> > ### Comment · Reviewer_7Se4 · 2021-09-02
> > **Thanks for the response**
> >
> > Thanks for the detailed response!
> >
> > On a separate note, I think it’d be interesting to design a task sequence for a second CL scenario that evaluates on different instances of a task. Meta-World is already designed so that we can sample multiple instances of each task type. This would be analogous to how CORe50 offers 3 continuous learning scenarios: new instances, new classes, and new instances + classes. Rather than introducing an entirely new type of task, e.g., “push-wall-v1” each time, the agent may need to solve a new instance of a previously seen type of task. Ideally, a lifelong learning agent should be able to solve completely new types of tasks as well as new instances of previously seen ones. It’s however just a suggestion and doesn’t affect my score.

---

> > > ### Author Response · Authors · 2021-09-03
> > > **Thanks for suggestion**
> > >
> > > Thank you for your suggestion. It is indeed interesting and might provide a more fine-grained measurement of forward transfer. We will consider it in the future version of our benchmark.
> > >
> > > As a side remark: in some experiments (not included in the paper), we considered semantically similar tasks (e.g., a task involving manipulation of dishes). It turned out that the current CL methods cannot achieve meaningful forward transfer in such cases (i.e., going above transfer between unrelated tasks).

---

### Official Review · Reviewer_g4u3 · 2021-07-19

**Rating:** 7
**Confidence:** 4

**Summary:**

A benchmark for continual reinforcement learning is provided, where the main task contains 20 subtasks. A certain number of choices are made such as the agent being aware of task change, not carrying forward the replay buffer, and learning policies with separate output heads per task. The evaluation was overall performance, forgetting, and forward transfer. Seven learning methods are compared extensively and results are provided.


**Main Review:**

The setup is not satisfying but it is also true that finding a continual learning benchmark acceptable is extremely hard. As a first start, I think this benchmark can be acceptable and an interesting place to start.

There is an emphasis that the different goals in continual learning are conflicting. However, that is neither a surprise nor unrealistic. Animals face the same tradeoffs, and these are exactly why continual learning is so challenging.

It seems the focus on computational and memory resource capacity is forgone. In the experiments and comparison, it was not taken into account. Different algorithms took different amounts of computations and memory, which were not described. If they took widely different amounts of computations, it is not clear that aligning their performance on a per time step basis for comparison is fair anymore.

In Table 1, all forgetting numbers are positive. Equation (3) suggests they should be negative. If that is not the case, then can you clarify how this forgetting number is calculated? And, does 0.73 for fine-tuning mean a high level of forgetting?

It is not clear what is meant by regularizing critic. It is also not clear whether the critic is being learned from scratch or not or whether it depends on the method.

*** post rebuttal ***

Thanks for the clarification.

**Time Spent Reviewing:**

4

---

> ### Author Response · Authors · 2021-08-10
> **Rebuttal answer to Rev g4u3**
>
> We thank you for the review.
>
> As to the _continual learning objectives being conflicting_, we agree that this is unsurprising and probably widely accepted by the community. What we are trying to argue (and we will make this claim clearer in the text) is that this conflict is still not sufficiently well researched by the AI/CL community (e.g., most works end up exclusively looking at catastrophic forgetting). We aim to increase awareness of this dilemma and provide an explicit benchmark to systematically look (at least) at the interplay between forward transfer and catastrophic forgetting. A new benchmark is needed because the settings used previously, like Atari, do not allow us to meaningfully look at the transfer question (as the relationship between tasks is quite limited).
>
> We acknowledge the information about the resources requirements is relevant and we will rectify this. Below, we present the table summarising time and memory. For memory, we report separately neural network parameters and external memory used. We normalize with respect to Fine-tuning (recall that this is a baseline not using any CL techniques). One can see that PackNet, the best in our comparison, is rather fast and has a relatively modest memory footprint. The table and an extended discussion will be included in the camera-ready.
>
>
> | Method | Normalized Running Time | Normalized Parameter Memory Overhead | Examples to remember | Comment |
> |---|---|---|---|---|
> | Fine-tuning | 1.00 | 1.0 | N/A | Baseline method - no CL techniques used. |
> | Reservoir | 2.20 | 1.0 | 1M | Reservoir uses a buffer of examples collected during training of the previous tasks. |
> | A-GEM | 1.47 | 1.0 | 20M | A-GEM uses a buffer of examples collected during training of the previous tasks. |
> | EWC | 1.13 | 2.0 | N/A | EWC uses parameters of the previous task & importance weights. |
> | L2 | 1.12 | 1.5 | N/A | L2 uses parameters of the previous task. |
> | MAS | 1.13 | 2.0 | N/A | MAS uses parameters of the previous task & importance weights. |
> | PackNet | 1.15 | 1.5 | N/A | PackNet uses masks assigning weights to tasks. |
> | VCL | 1.28 | 3.0 | N/A | Model is $2\times$ bigger because of Bayesian networks. VLC uses parameters of the previous task. |
>
> Note that for most of the methods we only need to keep the parameters of the actor (which is almost the same size as the critic).
>
> You are right about eq (3), it should be $F_i=p_i(t\cdot \Delta) - p_i(T)$. We double-checked that in our code, we use the correct definition; in particular, the values in Table 1 are confirmed. To be exact, small values denote (desirable) retention performance, while high indicate harmful forgetting. $0.73$ for fine-tuning is a very high forgetting value, rather unsurprising, as this (baseline) method does not attempt to mitigate forgetting. We will fix the error and clarify the description using the additional page available in the camera-ready version.
>
> Regarding critic learning: the critic is randomly initialized at the start of the first task and then is trained through the whole sequence, retaining information from previous tasks. That is, the weights are not reinitialized when changing a task. As for critic regularization, one may apply a specific continual learning method to both actor and critic (referred to as _regularized critic_) or only to actor, leaving the critic training unconstrained (e.g. apply the regularization penalty only to actor’s weights in case of L2, EWC or MAS). We use critic regularization only in replay-based methods (for which it does not make sense not to regularize the critic). We will provide a clarification in the camera-ready version.

---

### Decision · Program_Chairs · 2021-09-27

**Decision:**

Accept (Poster)

**Comment:**

I think reviewers were generally satisfied that this was a useful contribution.  I found the reviews generally fair and the paper itself generally meritorious.  Obviously there are some concerns, mostly related to the inherent difficulty of producing a satisfactory continual learning benchmark.  This paper is somewhere near the current pareto-frontier of the tradeoff between "doing something useful but hard" and "doing something in a completely satisfactory way".   I don't completely know if that frontier is close enough to the absolute quality threshold for NeurIPS, but I tend to think it is.